# Genomic disparities between cancers in adolescent and young adults and in older adults

**Xiaojing Wang** [1,2,3] ✉, **Anne-Marie Langevin**[3,4], **Peter J. Houghton**[1,3,5] & **Siyuan Zheng** [1,2,3] ✉

Cancers cause significant mortality and morbidity in adolescents and young adults (AYAs), but their biological underpinnings are incompletely understood. Here, we analyze clinical and genomic disparities between AYAs and older adults (OAs) in more than 100,000 cancer patients. We find significant differences in clinical presentation between AYAs and OAs, including sex, metastasis rates, race and ethnicity, and cancer histology. In most cancer types, AYA tumors show lower mutation burden and less genome instability. Accordingly, most cancer genes show less mutations and copy number changes in AYAs, including the noncoding *TERT* promoter mutations. However, *CTNNB1* and *BRAF* mutations are consistently overrepresented in AYAs across multiple cancer types. AYA tumors also exhibit more driver gene fusions that are frequently observed in pediatric cancers. We find that histology is an important contributor to genetic disparities between AYAs and OAs. Mutational signature analysis of hypermutators shows stronger endogenous mutational processes such as MMR-deficiency but weaker exogenous processes such as tobacco exposure in AYAs. Finally, we demonstrate a panoramic view of clinically actionable genetic events in AYA tumors.

Cancer affects ~89,000 adolescents and young adults (AYAs, ages 15–39) annually in the US[1]. This incidence is about 5% of that in older adults (OAs, ages > 39) but eight times higher than in children ages 0–14. In the last 30 years, survival rates for AYA patients have shown little improvement while significant improvements have been made for children and older adults[2–4]. For example, the 5-year survival rate for acute lymphocytic leukemia is 60% in AYAs compared with 91% in children. Many clinical centers group AYA patients with either younger children or older adults due to the lack of specialized AYA cancer programs. AYA patients are also less enrolled in clinical trials and are often diagnosed at advanced stages[5–7].

Several studies have documented genetic and clinical differences between AYAs and OAs. In breast cancer (BRCA), AYAs are more likely to present with aggressive subtypes and advanced diseases[8,9]. In melanoma, AYAs show more *BRAF* mutations[10,11]. Despite these important findings, AYA associated genomic alterations are still poorly understood. A systematic comparison of AYA and OA cancers may provide insights in AYA cancers but is currently lacking. Resources such as The Cancer Genome Atlas (TCGA) and International Cancer Genome Consortium have been recently used to study age-associated genomic alterations[12–15]. However, both resources are focused on adult cancers and have few samples from AYAs. More importantly, cancer types included in these studies often do not reflect common cancers observed in AYAs.

In this study, we use cancer panel sequencing data of more than 100,000 tumors from AACR GENIE(Genomics Evidence Neoplasia

[1]Greehey Children's Cancer Research Institute, UT Health San Antonio, San Antonio, TX, USA. [2]Department of Population Health Sciences, UT Health San Antonio, San Antonio, TX, USA. [3]MD Anderson Mays Cancer Center, UT Health San Antonio, San Antonio, TX, USA. [4]Department of Pediatrics, UT Health San Antonio, San Antonio, TX, USA. [5]Department of Molecular Medicine, UT Health San Antonio, San Antonio, TX, USA. ✉e-mail: wangx11@uthscsa.edu; zhengs3@uthscsa.edu

Information Exchange)[16] to investigate the genetic and clinical disparities between AYAs and OAs with cancer. We identify AYAs as individuals aged 15–39 per guidelines from National Cancer Institute and the Adolescent and Young Adult Oncology Progress Review Group. Patients aged >39 were considered OAs.

## Results

### Clinical disparities between AYAs and OAs with cancer

AYA samples (ages 15–39) account for 8.1% ($n = 9140$) of the GENIE cohort (v 9.0). We excluded cases without precise age information (including '<18' and 'unknown') from our study. We first examined clinical and epidemiological characteristics of cancers in AYAs using 19 cancer types that had more than 100 cases in both AYAs and OAs (Supplementary Fig. 1). This cohort (AYA: $n = 7579$; OA: $n = 72,491$) comprised cancer types commonly observed in AYAs including thyroid cancer (THCA), BRCA, testicular germ cell tumors, soft tissue sarcoma, melanoma, and glioma. As a biological attribute, both sexes were represented in the cohort except in sex-associated cancer types including BRCA, ovarian cancer, and cervical cancer (CESC). No sex related filters were applied to ensure that the cohort reflects the clinical representation of both sexes.

We found a significant increase in females in AYA THCA (69% in AYAs vs. 49% in OAs; percentages are listed in the same order throughout the work), renal cell carcinoma (48% vs. 27%), melanoma (52% vs. 38%), head and neck cancer (41% vs. 25%), esophagogastric cancer (39% vs. 28%) and colorectal cancer (49% vs. 45%) (FDR < 0.05; Fig. 1a, Supplementary Data 1), suggesting sex is an important etiological factor in AYA cancers. No cancer type showed significantly higher ratios of male patients in AYAs than in OAs.

Proportions of patients diagnosed with metastatic diseases differed between the two groups in eight of the 19 cancer types (Fig. 1b, FDR < 0.05). Among the eight, six showed less metastatic cases in AYAs. Only non-small cell lung cancer (49% vs. 37%) and pancreatic cancer (50% vs. 39%) showed higher proportions of metastatic patients in AYAs, indicating possible delays in diagnosis for these two deadly cancer types.

We observed a significant increase in non-Hispanic Asian patients in nine of the 19 cancer types in AYAs (Fig. 1c, FDR < 0.05); on average, the proportion of non-Hispanic Asian in AYAs was 1.8-fold higher than in OAs. In eight cancer types, the proportions of Hispanics were higher in AYAs (Fig. 1c, FDR < 0.05). In particular, the likelihood ratio in leukemia was more than five times higher for Hispanic AYAs than OAs (FDR = 1.4e−05). The proportion of Non-Hispanic Black was generally similar between AYAs and OAs (average likelihood ratio 1.28) except in soft tissue sarcoma and renal cell carcinoma (Fig. 1c). In soft tissue sarcoma, the proportion of non-Hispanic Black was 7.8% in AYAs compared to 3.9% in OAs (FDR = 0.001). In renal cell carcinoma, these numbers were 13.5% and 3.7% (FDR = 2.7e−05). This result confirms previous studies that reported overrepresentation of younger Black patients in renal cell carcinoma[17,18]. In 13 cancer types, the proportions of non-Hispanic White were lower in AYAs than in OAs (average likelihood ratio 0.82). This is expected because of the increase in other ethnic groups. Details including the proportions of each ethnic group in every cancer type is summarized in Supplementary Data 1. These results demonstrate a distinct fabric of patient ethnicity in AYA patients.

We next compared histological subtypes between AYAs and OAs. In the GENIE dataset, the 19 cancer types were further divided into 371 subtypes. Some of these subtypes were redundant because contributing institutes use inconsistent nomenclature to describe the same entities, e.g., glioblastoma and glioblastoma multiforme. With this caveat, we found 105 differentially distributed histological subtypes (FDR < 0.05), 67 of which were overrepresented in AYAs (Fig. 1d, Supplementary Data 1). In every cancer type we examined, at least two histological subtypes showed differential distribution, indicating a

widespread disparity in cancer histology between AYAs and OAs. We then compared molecular subtypes from TCGA[19] in nine cancer types where we could find at least 20 AYA samples. We observed 10 differentially distributed subtypes between AYAs and OAs (FDR < 0.05), eight of which were from glioma (Supplementary Data 1). The other two subtypes were 'BRAF_Hotspot_mutants' in melanoma and 'pseudohypoxia' in pheochromocytoma and paraganglioma. Notably, although previous studies observed more triple-negative samples in AYA BRCA[9,20], we did not find this enrichment.

### Multiple factors contribute to genomic disparities between AYAs and OAs

To characterize genetic disparities between AYA and OA cancers, we first compared the overall tumor mutation burden (TMB), a feature closely associated with age. We focused on cancer types that had more than 100 samples in each age group and further required each sample sequenced by panels that cover more than 0.9 Mb of exonic regions. This led to a total of 52,919 samples (AYA, $n = 5295$; OA, $n = 47,624$) across 16 cancer types. Previous studies suggested that panels with lower coverages are not reliable for inferring TMB[21–24]. The median TMBs in AYAs were 1.7-fold lower than in OAs. We observed significant differences in 13 cancer types, and all showed lower TMBs in AYAs (FDR < 0.05, Wilcoxon rank sum test; Fig. 2a and Supplementary Data 2). The three cancer types that were not significant were bone cancer (FDR = 0.39), CESC (FDR = 0.06) and glioma (FDR = 0.54).

We next compared mutation rates of cancer genes. Because GENIE gene panels are designed by numerous contributing institutions, directly merging samples from these panels would increase sample size at the cost of overlapping genes. To balance this tradeoff, we adopted a single gene test strategy to maximize sample sizes for each comparison. Briefly, for each gene, we aggregated samples from any panel if it covers the whole exonic region of the gene. With this approach, each comparison uses a slightly different pool of samples, but the large sample size allows sensitive detection of differences. In total, we curated 1029 genes and 83,482 samples from 21 panels.

We used Fisher's exact test to compare mutation rates between AYAs and OAs followed by multiple hypothesis testing. The univariable approach allowed us to use the full GENIE dataset but could not adjust for confounders. To complement Fisher's exact test, we used multiple logistic regression to control for confounders. More discussions on the two models were provided in Method. We constructed a backbone model by controlling for histological subtype, metastatic status, and patient sex based on the clinical data analysis (Fig. 1). Other potential confounders included common clinical variables such as tumor stage and patient ethnicity, and cancer type specific variables such as alcohol consumption for liver cancer or smoking history for lung cancer. Most of these clinical variables were not available in GENIE. Molecular variables such as TMB can also confound mutation rates, because older patients may accumulate more mutations than younger patients due to ageing. Furthermore, GENIE data were contributed by multiple institutions; thus, sample site could be another confounder.

To evaluate the impact of these confounders, we calculated the AYA effect size with and without a confounder in the backbone model. Significant changes in effect size indicate a need to adjust for the confounder in the model. We found that adding tumor stage, patient ethnicity, and sample site to the backbone model caused little change to AYA effect sizes (Supplementary Fig. 2; Method). However, including TMB led to more significant genes that showed a higher mutation rate in AYAs ($n = 30$) (Supplementary Fig. 2, Supplementary Data 3). Notably, 77% ($n = 23$) of the genes were mutated in <5 AYA samples at an average mutation rate of 5% in AYAs. Estimates of AYA effects for these genes also showed larger error margins, a strong indication that adding more variables to the model reduces its statistical power. On the other hand, the TMB model excluded *MSH2* in colorectal cancer as an AYA enriched gene by attributing its higher mutation rate to TMB effect.

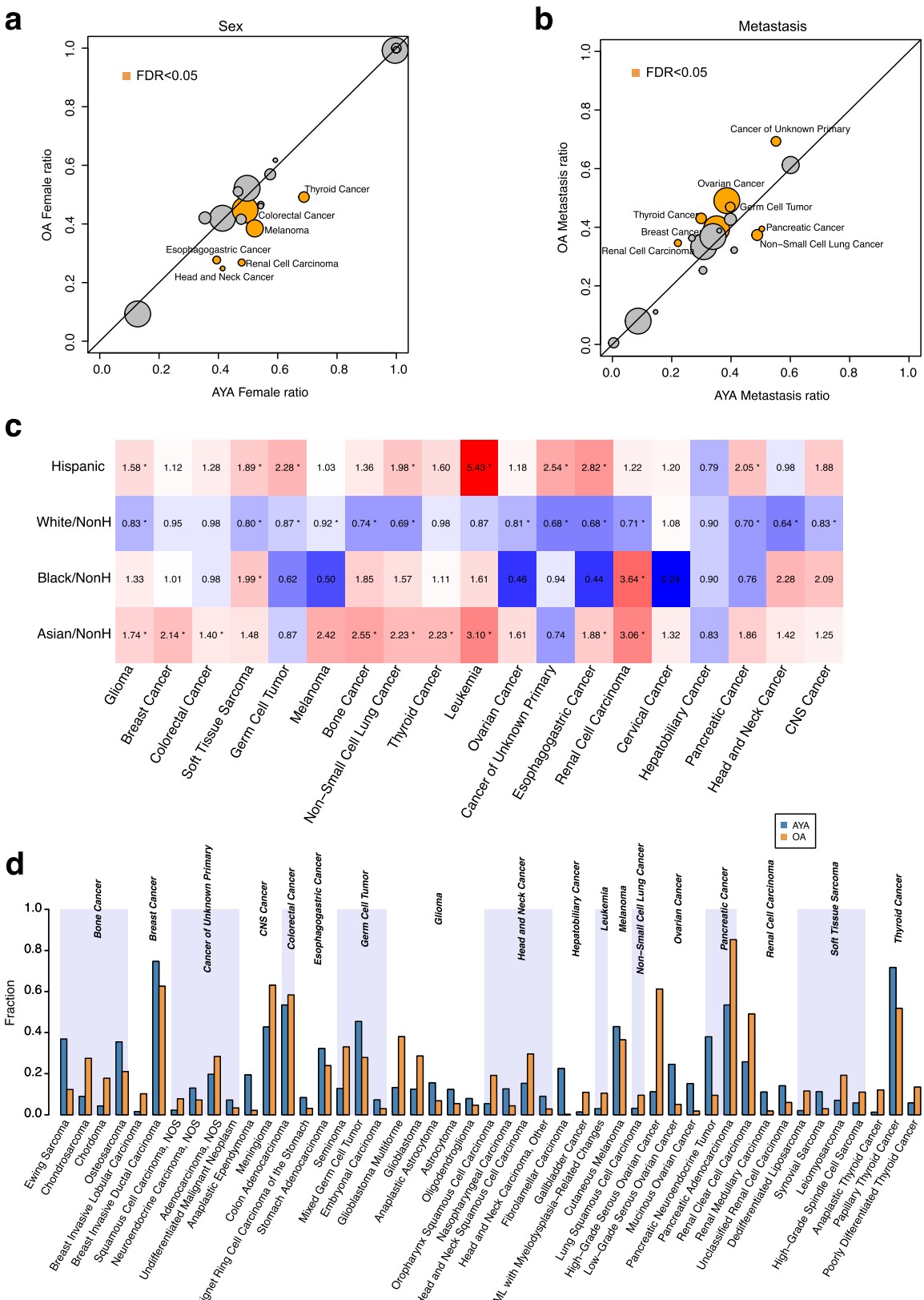

**Fig. 1 | Clinical disparity between AYAs and OAs. a** Female fraction, (**b**) and metastasis fraction between AYA and OA in 19 tumor types. Circle size corresponds to AYA sample counts of the specific tumor types. Orange color indicates statistically significant differences (FDR < 0.05). Cancer types were labeled for those with significant differences (FDR < 0.05). **c** Comparison of race and ethnicity between AYA and OA samples. Color depth denotes the average likelihood ratio. Red, >0; blue, <0. Superscript stars indicate the difference were statistically significant (FDR < 0.05). **d** Comparison of histological subtype between AYAs (blue) and OAs (orange). Only major subtypes (≥10%) with statistically significant differences (FDR < 0.05) were include in the plot. Source data are provided as a Source Data file.

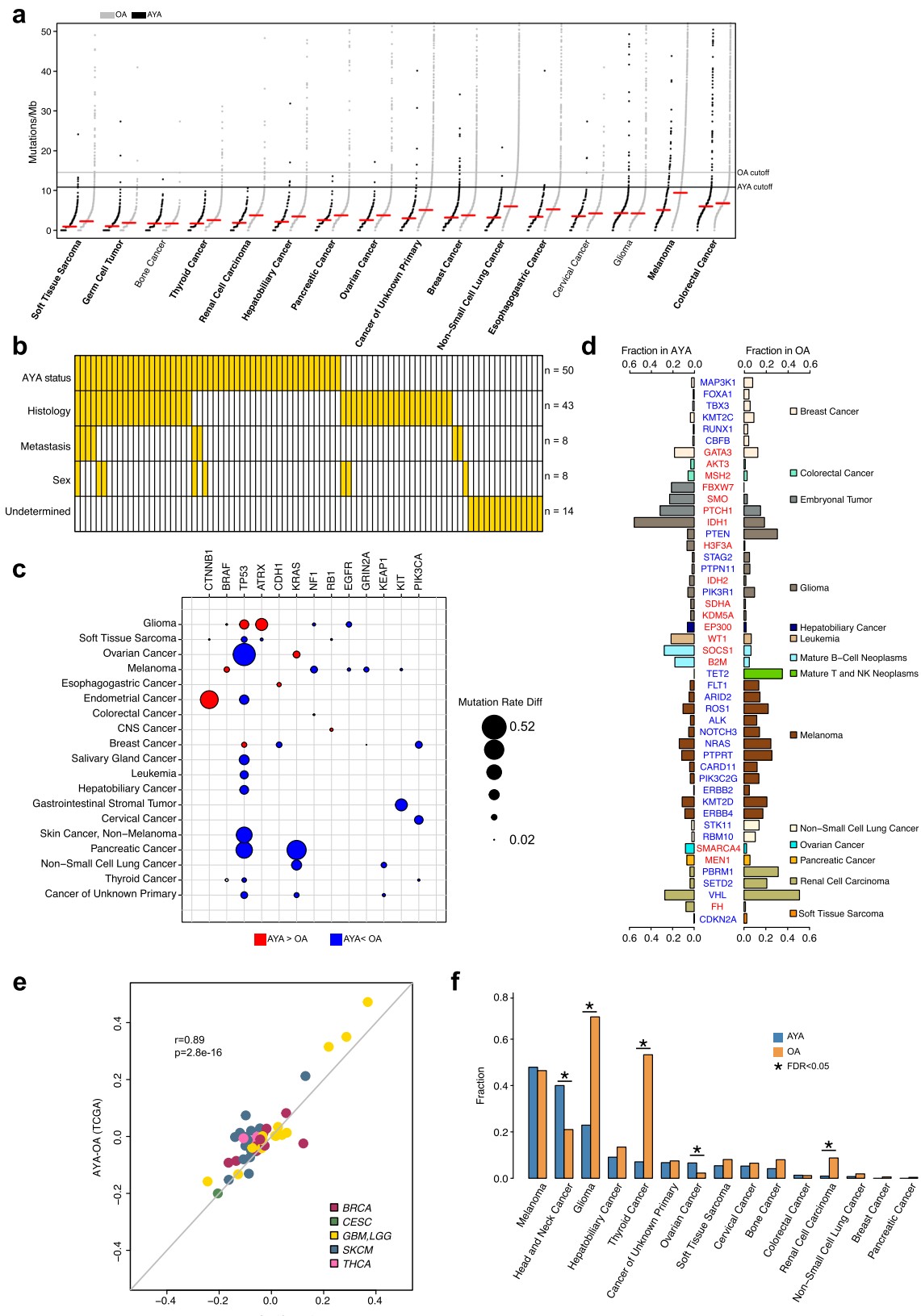

This was unwanted, because *MSH2* mutations cause DNA repair deficiency and high TMB. More discussions on confounder adjustments, including strengths and limitations of TMB adjustment, can be found in Method. Detailed comparisons are provided in Supplementary Data 3. Raw outputs of the models were provided in Supplementary Data 4. We also tested the impact of metastasis by comparing our original results

and results based on primary tumors only (Method). The two approaches yielded consistent results (Supplementary Fig 3), suggesting the original model could properly control for metastasis related effects. For better coherence, we mainly described results from the Fisher's test and backbone model in the work. But we also noted results from the TMB adjustment when appropriate.

**Fig. 2 | Mutation disparity between AYAs and OAs. a** Mutations (including non-silent substitutions and small insertions/deletions) per coding megabase (Mbs) in AYAs and OAs. Tumor types are ordered by the median mutation rate (red line) in AYAs. Cancer types that show statistical significance (FDR < 0.05) are highlighted in bold. Black and gray lines represent two cutoffs we used to select hypermutators (AYA cutoff: 10.83 mutations/Mb; OA cutoff: 14.56 mutations/Mb). **b** A summary of clinical factors that underlie differential mutational rates per logistic regression models. **c** Recurrent genes (*n* = 13) with different mutational rates (FDR < 0.05). Circle size represent the absolute value of mutation rates differences (AYA-OA).

Red indicates higher in AYAs, and blue indicates higher in OAs. **d** 47 genes with differential mutational rates (FDR < 0.05) in a specific cancer type. Genes with higher mutational rates in AYA are in red, otherwise in blue. **e** Differences in mutation rates between AYAs and OAs based on GENIE (*x* axis) and TCGA (*y* axis) cohorts. Each dot represents one gene-cancer pair that was identified showing different mutational rate between AYA and OA in GENIE data. **f** TERTp mutational rates in AYAs (blue) and OAs (orange) in 15 tumor types. Asterisk indicates statistically significant difference (FDR < 0.05). Source data are provided as a Source Data file.

We identified 79 significant gene-cancer pairs using Fisher's test and 50 pairs using the backbone logistic model (FDR < 0.05). The two methods identified 88 pairs when taking the union, among which were 41 common pairs identified by both (Supplementary Data 3).

To understand what contributes to the genetic disparity, we analyzed each pair for factors that explain its significance (Fig. 2b). Among the 50 pairs identified by the logistic model, 25 were solely explained by patient AYA status. For the rest (*n* = 25), one or multiple factors in addition to the AYA status were significantly associated with mutation rates, including histological subtype (*n* = 22), sex (*n* = 5), and metastatic status (*n* = 6). For the 38 pairs that were identified only by Fisher's exact test, we similarly examined their significance in the logistic model. We found that 24 could be explained by histology (*n* = 21), sex (*n* = 3), and metastatic status (*n* = 2). The remaining 14 pairs did not reach statistical significance for any of the confounders included despite their overall significance by the Fisher's exact test. These genes typically had low mutation rates or were found in cancer types with small AYA cohorts. Taken together, these results show that the histological subtype contributes significantly to the cancer genetic disparities between AYAs and OAs.

We use *CDH1* and *SMARCA4* as examples to demonstrate how histological subtype can impact mutation rates. *CDH1*, a membrane cadherin, was mutated in 19% of AYAs but only in 9% of OAs with esophagogastric cancer. This cancer type can be divided into stomach and non-stomach subtypes. Non-stomach cancers had few *CDH1* mutations (3%). Among AYA patients, 53% had stomach cancer, of which 31% carried a *CDH1* mutation. In comparison, 32% of OA patients had stomach cancer, of which 21% carried a *CDH1* mutation. Thus, higher fraction of the stomach cancer subtype combined with its increased mutation rate led to an overall *CDH1* mutation rate twice as high in AYAs as in OAs (19% vs. 9%). In ovarian cancer, *SMARCA4* mutations are a defining molecular feature of small cell ovarian cancer[25] and are observed in 92% of AYA patients diagnosed with this subtype. The more common ovarian cancer subtype, high-grade serous ovarian cancer, rarely harbor *SMARCA4* mutations (<1% in both age groups). The AYA and OA patients had drastically different representations of the two subtypes: 8% of AYAs were diagnosed with small cell subtype vs. 0.3% of OAs; 11% of AYAs were diagnosed with high-grade serous subtype vs. 61% of OAs. This difference led to an overall *SMARCA4* mutation rate of 8% in AYAs and 2% in OAs with ovarian cancer.

## Cancer genes that exhibit different mutation rates between AYAs and OAs

Because histological subtype and other clinical variables are integral to clinical manifestation, we focused on 88 gene-cancer pairs in the following analyses. We divided the significant genes into recurrent if they were found in two or more cancer types (*n* = 13; Fig. 2c), and non-recurrent otherwise (*n* = 47; Fig. 2d).

Most recurrent genes showed a lower mutation rate in AYAs (Fig. 2c), consistent with AYAs' overall lower TMBs. The most frequently observed gene was *TP53*, which showed significantly lower mutation rates in 10 cancer types and higher rates in only two cancer types (BRCA and glioma). Controlling for TMB additionally identified

higher *TP53* mutation rate in AYA appendiceal cancer. RTK/PI3K pathway genes including *KIT, EGFR, NF1*, and *PIK3CA* showed lower mutation rates in AYAs, thus targeting this pathway will likely make less clinical impact to AYAs. *BRAF* and *CTNNB1*, two oncogenes frequently mutated in childhood cancers, showed higher mutation rates in AYAs. Interestingly, *KRAS*, the gene upstream to *BRAF* in the RAS/MAPK pathway, showed a predominantly lower mutation rate pattern, suggesting genes from the same pathways can be preferentially selected during tumorigenesis in different age contexts.

The non-recurrent genes were listed in Fig. 2d. Genes from the RTK/PI3K pathway, including *PTEN, PIK3R1* in glioma, and *ALK, PIK3C2G, ERBB4* in melanoma, showed lower mutation rates in AYAs, reiterating our earlier observation on recurrent genes. Notably, epigenetic modifiers frequently showed more mutations in AYAs, including *H3F3A, KDM5A* and *IDH1/2* in glioma, *EP300* in liver cancer, *SMARCA4* in ovarian cancer, and *MEN1* in pancreatic cancer. In AYA BRCA we identified more frequent *FANCM, FANCD2*, and *BRCA1* mutations when controlling for TMB, suggesting the role of DNA damage repair in this cancer type.

Because some genes with frequent mutations in pediatric cancers such as *BRAF* and *H3F3A* also showed higher mutation rates in AYAs than in OAs, we tested if their increased mutation rates were an extension of their frequent mutations in pediatric cancer. We used glioma as an example, because it has the most pediatric samples (age 0–18 y.o.) in GENIE. Among the five frequently mutated glioma genes (*BRAF, IDH1, TP53, IDH1*, and *H3F3A*), *BRAF* and *H3F3A* mutation rates continuously decreased as patient ages increased from pediatric to AYAs and to OAs. However, *TP53, ATRX* and *IDH1* defied this pattern, with all three showing the highest mutation rates in AYAs. These data suggest that genetically the AYA group is not simply an extension of pediatric tumors.

We next used TCGA datasets to validate our results. We required a minimum of 50 AYAs and OAs in a TCGA cancer type. This led to 5 cancer types, THCA, BRCA, glioma (GBM/LGG), CESC, and melanoma (SKCM). We then compared mutation rate differences between OAs and AYAs for the differential genes we identified in GENIE. The differences were reproduced using TCGA data (*r* = 0.89, *p* = 2.8e−16; Fig. 2e and Supplementary Fig. 4), supporting the robustness of our results.

## TERT promoter mutation in AYAs

Active telomerase is critical for maintaining cancer cell immortality. Telomerase consists of an enzymatic subunit encoded by *TERT* and an RNA template encoded by *TERC*. *TERT* expression is usually suppressed in normal somatic cells. For tumors that arise from a telomerase incompetent cell, genetic alterations such as *TERT* promoter (TERTp) mutations reactivate *TERT* expression, leading to active telomerase. Tumors may also arise from stem cells that are telomerase competent. For these cells, TERTp mutations provide few evolutionary advantages. We reasoned that cancers in AYAs are more likely to originate from telomerase competent cells because of the younger patient age, and thus, may carry less TERTp mutations. To test this hypothesis, we examined TERTp hotspot mutations that occur at 124 and 146 base pairs upstream from its translation start site.

We compared TERTp mutations between AYAs and OAs in 15 cancer types using 12 panels that covered this region. In AYAs, melanoma had the highest TERTp mutation rate (47.9%), followed by head and neck cancer (40.0%), glioma (22.9%) and hepatobiliary cancer (9.1%) (Fig. 2f, Supplementary Data 5). These tumor types also exhibit frequent TERTp mutations in OAs, suggesting cancers in AYAs adopt similar pathways to immortality as in OAs.

Of the 15 cancer types we tested, five showed significant differences (FDR < 0.05), including glioma (22.9% vs. 69.6%), THCA (7.1% vs. 53.3%) renal cell carcinoma (0.9% vs. 8.8%), head and neck cancer (40.0% vs. 21.0%), and ovarian cancer (6.6% vs. 2.2%). This pattern persisted when we split AYAs into a younger (aged < 30) and an older group (aged 30–39) (Supplementary Fig. 5). Interestingly, AYA gliomas had more mutations in *ATRX* (41.3% vs. 12.5%), a chromatin regulator previously associated with Alternative Lengthening of Telomeres[26] (ALT), a telomere maintenance pathway complementary to active telomerase. For THCA, logistic regression model analysis suggests AYA status ($p = 3.6e{-}12$; OR = 0.07, 95% CI 0.03–1.14), metastasis ($p = 1.1e{-}07$; OR = 2.26, 95% CI 1.68–3.07), and histological subtypes all contributed to the disparity in TERTp mutations.

To understand why TERTp mutations were nearly two-fold more frequent in AYA head and neck cancer than in OAs, we constructed a logistic regression model considering AYA status, sex, histological subtype, and metastatic status. We found both AYA status ($p = 0.01$; OR = 2.67, 95% CI 1.71–4.24) and sex ($p = 0.001$; male, OR = 0.52, 95% CI 0.36–0.71) were significantly associated with TERTp mutation.

### Gene level CNA

To measure the global pattern of copy number alterations, we calculated a genomic instability (GI) score as the proportion of the genome affected by copy number changes. GI scores were significantly different between AYAs and OAs in nine of the 16 cancer types examined (FDR < 0.05, Wilcoxon rank sum test; Fig. 3a, Supplementary Data 6); all but BRCA showed lower scores in AYAs (BRCA median 0.26 in AYAs vs. 0.23 in OAs), suggesting AYAs generally have lower GI.

To understand what drives higher GI in AYAs with BRCA, we compared molecular subtypes between AYAs and OAs using the TCGA BRCA cohort, and we did not observe differences (Supplementary Data 1, Basal, Her2, LumA, LumB, all FDR > 0.7). Metastasis was previously associated with increased GI[27]. To examine its impact on GI scores, we included metastasis in a multivariate regression model. We found metastasis was indeed significantly associated with higher GI score. However, metastasis could not explain the increased GI in AYA BRCA, because AYAs contained less metastatic samples (40% vs. 49% in OA). To further validate this, we limited the comparison to only primary tumors and still found higher GI scores in AYAs ($p = 0.005$, Wilcox rank sum test, Supplementary Fig. 6a). Correlating with higher GI, the mutation rate of *TP53* was also higher in AYAs (52.6% vs. 40.4%).

We next compared gene level copy number changes between AYAs and OAs and identified 46 significant gene-cancer pairs across 9 cancer types, including 32 amplifications and 14 deletions (Fig. 3b, Supplementary Data 6). Most of the pairs were found in glioma ($n = 18$), BRCA ($n = 13$) and soft tissue sarcoma ($n = 6$). Amplifications were more frequent in AYAs ($n = 21, 66\%$) whereas deletions were more frequent in OAs ($n = 11, 79\%$).

The genes identified in copy number comparisons can be largely grouped into p53, cell cycle, RTK, and epigenetic modification pathways, echoing our mutation analysis. For the p53 pathway, we identified *MDM2, MDM4,* and *RB1* in multiple cancer types. For the cell cycle pathway, we identified *CDK4, CCNE1, CCND1, CCND2,* and *CDKN2A/B*. Interestingly, the two mesenchymal cancer types—glioma and sarcoma—shared several common alterations including deletions of *CDKN2A/B* and amplifications of *MDM2* and *CDK4*. In both cancer types, these alterations were less frequent in AYAs. RTK/PI3K pathway genes, including *EGFR, PDGFRA,* and *PTEN* showed less copy number

alterations in AYAs. Finally, we observed more deletions of a chromatin modifier, *ATRX*, in AYA gliomas where *ATRX* deletions and mutations were mutually exclusive, suggesting this gene is haplo-insufficient. Loss of *ATRX* has been associated with the telomere maintenance pathway ALT. Interestingly, we observed increased amplification of *RTEL1*, a helicase involved in telomere maintenance in AYA BRCAs (6% in AYAs vs. 3% in OAs).

More than half of the amplified genes overrepresented in AYAs (12/21) were identified in BRCA. Amongst the 12 genes, eight were located at chromosome 17q12-24, including *ERBB2, CDK12, HOXB13, SPOP, PPM1D, CD79B, BRIP1* and *PRKAR1A*. These genes are located in two amplicons, one encompassing *ERBB2* and *CDK12* (17q12) and the other encompassing the remaining genes. The first amplicon co-occurred with *TP53* mutations, whereas the second co-occurred with *GATA3* mutations (Fig. 3c). Both *TP53* and *GATA3* mutations were enriched in AYAs. Interestingly, the two groups of events were mutually exclusive, and *TP53* mutant AYA samples showed higher GI scores than *GATA3* mutant AYA tumors ($p = 0.0001$, Wilcox rank sum test, Supplementary Fig. 6b). *GATA3* is a transcription factor that regulates tissue development and immune responses[28,29]. These data implicate two distinct regulatory programs underlying AYA BRCA.

### Gene fusions comparison between AYA and OA

Twenty-one assays in GENIE can detect gene fusions. To minimize technical batch effects due to design differences, we limited our analysis to 9564 fusions called by the MSK-IMPACT468 panel.

We first examined differences in recurrent fusions, which were defined as being detected in more than 1% of a cancer cohort. At FDR 0.05, we identified six differential fusion-cancer pairs (Fig. 3d, Supplementary Data 7), five of which were more frequent in AYAs, including WT1-EWSR1 (9.2% vs. 0.4%) and EWSR1-ATF1 (2.9% vs. 0.6%) in soft tissue sarcoma, EML4-ALK (16.4% vs. 1.5%) in non-small cell lung cancer, RET-NCOA4 (8.2% vs. 0%) in THCA, and DNAJB1-PRKACA (16.3% vs. 0%) in hepatobiliary cancer. To evaluate the transcriptional consequences of these fusions, we compared expression level of the oncogenes involved in the fusion between fusion-positive and fusion-negative tumors using TCGA data. The results show that the oncogenes were highly expressed in fusion-positive tumors (Supplementary Fig. 7). The only fusion with a higher prevalence in OAs was intragenic fusions of EGFR in glioma. *EGFR* amplifications and mutations were also more frequent in OA gliomas. Fusions between *EWSR1* and partner genes often define pathological subtypes of sarcoma, and thus, their higher incidences in AYAs reflect the higher incidences of the corresponding subtypes. Specifically, EWSR1-WT1 fusion is a defining molecular feature of desmoplastic small round cell tumor, and EWSR1-ATF1 defines clear cell sarcoma. Both sarcoma subtypes showed a higher incidence in AYAs. Similarly, the DNAJB1-PRKACA fusion was observed in more than 90% of fibrolamellar liver carcinoma[30], which accounted for 22.5% of liver cancers in AYAs but only 0.1% in OAs.

The EML4-ALK fusion was found in 10 AYAs with non-small cell lung cancer. This fusion was previously associated with female and non-smokers[31]. In our study, eight of the 10 AYA cases were female and nine were diagnosed with metastatic disease. Nevertheless, the increased frequency of EML4-ALK fusions in AYAs suggest *ALK* inhibitors may be clinically important for this group. *RET* fusions were previously associated with radiation in THCA[32]. Consistently, we found most RET-NCOA4 fusions in post-treatment metastatic papillary THCA. We did not find any RET-NCOA4 fusions in the OA thyroid tumors ($n = 412$).

Since only a few recurrent fusions were found due to their rarity, we expanded our analysis to fusion partner genes. We limited our analysis to genes with a minimum occurrence of 2% in each cancer cohort. We identified 16 gene-cancer pairs at FDR threshold 0.05 (Fig. 3e, Supplementary Data 7), and eight were from the recurrent

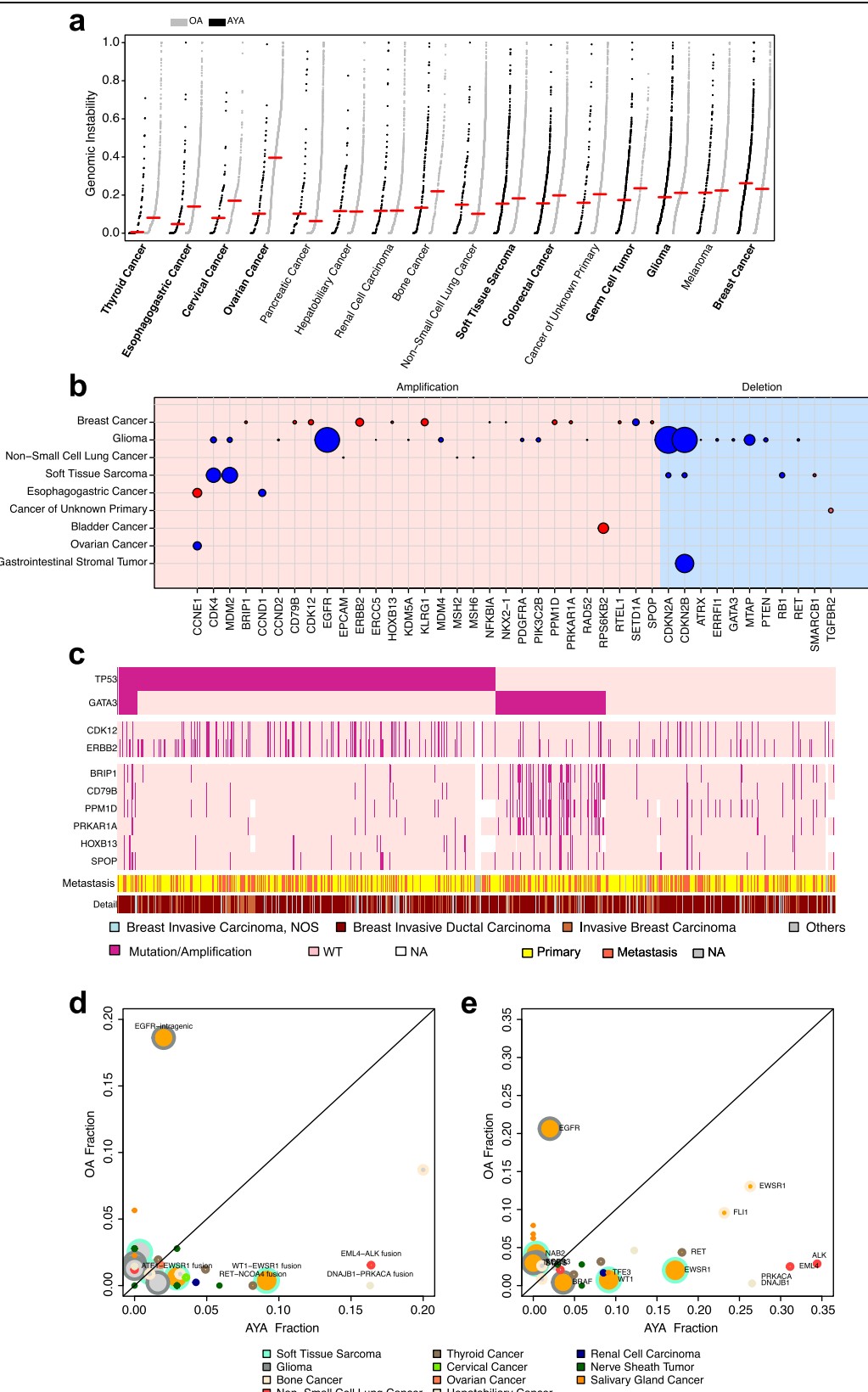

**Fig. 3 | CNVs and fusions in AYAs. a** Genomic instability (GI) scores in AYAs and OAs. Cancer types are ordered by median GI score (red line) in AYAs. Cancer types in bold font are statistically significant (FDR < 0.05). **b** 46 gene level CNAs (left, amplification; right, deletion) that showed statistically different frequencies (FDR < 0.05) between AYAs and OAs. Circle size represents differences in alteration rate (AYA-OA). Red indicates the CNV is more common in AYAs, and blue indicates otherwise. **c** Two amplicons in AYA breast cancer and their co-occurrence with *TP53* and *GATA3* mutations. **d**–**e** Comparison of gene fusions (**d**) and fusion involved genes (**e**) between AYAs and OAs. Circle size corresponds to AYA sample counts of the specific tumor types. Circles filled with orange color indicate statistically significant differences (FDR < 0.05). The colors of circle border represent tumor types. Source data are provided as a Source Data file.

fusions aforementioned. Among the eight pairs, four were more frequent in AYAs.

These four gene-cancer pairs were *EWSR1* (26.3% vs. 13.0%) and *FLI1* in bone cancer (23.2% vs. 9.6%), *BRAF* in glioma (3.6% vs. 0.4%), and *TEF3* in renal cell carcinoma (8.5% vs. 1.7%). Again, the increased presence of these fusion genes can be explained by enrichment of fusion-driven cancer subtypes in AYAs. For instance, fusions involving *EWSR1* and *FLI1* drive Ewing sarcoma, which makes up 36.9% of bone cancers in AYAs compared to 12.3% in OAs. The *BRAF* fusions were largely attributable to pediatric pilocytic astrocytoma. *TEF3* fusions were a marker of translocation renal cell carcinoma, a subtype observed in 7.4% of AYA renal cell carcinomas but only in 1.3% of OAs. Notably these cancer types are commonly observed in children; thus, fusion-targeting strategies developed for childhood cancers can benefit a substantial proportion of AYA patients.

The histological subtype does not always explain fusion enrichment. With borderline significance (FDR = 0.06), *FGFR2* fusions were more prevalent in AYA hepatobiliary cancer (12.2% vs. 4.6%). Eighty-four percent of the *FGFR2* fusions were found in the intrahepatic cholangiocarcinoma, a subtype that made up 37% of OAs and 31% of AYAs. Within the subtype, 27% of AYAs harbored a *FGFR2* fusion compared to only 11% of OAs.

Four genes showed a higher fusion rate in OAs included *NAB2* (0.4% vs. 4.2%) and *STAT6* (0.4% vs. 2.9%) in soft tissue sarcoma and *FGFR3* (0.4% vs. 3.1%) and *TACC3* (0 vs. 3.0%) in glioma. These fusions generally occur at low frequencies (Fig. 3e).

## Hypermutated samples in AYAs and implications for immunotherapy

Immunotherapies including immune checkpoint blockade (ICB) can induce durable tumor regression in some patients. To examine how ICB may benefit AYA patients, we compared the overall survival of patients who have received immune-checkpoint inhibitors (ICI) using the MSK immunotherapy cohort ($n = 1647$)[33]. Across pan-cancer, AYA showed worse overall survival ($p = 0.026$; HR = 1.328, 95% CI: 1.032–1.708) (Fig. 4a). A multivariate analysis controlling for cancer type, metastatic status, sex confirmed the association between worse survival and AYAs with borderline significance ($p = 0.090$; HR = 1.251, 95% CI: 0.965–1.622). This pattern was corroborated when we repeated the analysis in individual cancer types; though most cancer types did not reach statistical significance except renal cell carcinoma ($p = 0.0077$), they showed the trend toward worse survival in AYAs (Fig. 4b).

The worse responses to ICIs by AYAs can be partly explained by their lower tumor mutation load (TMB). We reasoned that AYA patients with higher TMB may respond better. To provide an overview of such cases, we examined hypermutators in AYAs. The hypermutator phenotype has been studied extensively in adult and childhood cancers[34,35] but not in AYAs. To identify hypermutators, we calculated mutation density, i.e., the number of coding nonsynonymous mutations per Mb, for 6540 AYAs and 65,431 OAs. Tumors with outlier mutation density in each age group were considered hypermutators (Methods; cutoff AYA, 10.83 mut/Mb; OA, 14.56 mut/Mb). We identified 340 hypermutators in AYAs (5.2%) and 5632 hypermutators in OAs (8.6%). We did not observe significant differences in the proportions of hypermutators between AYAs and OAs except in non-small cell lung cancer (Supplementary Data 8, 1.8% in AYAs vs. 11.7% in OAs), melanoma (20.9% vs. 36.4%) and colorectal cancer (16.9% vs. 10.2%).

To gain a mechanistic understanding of hypermutators in AYAs, we assigned their mutations to COSMIC cancer signatures (version 2.0) using deconstructSigs[36]. In total, we identified a dominant signature (signature score > 0.4) in 92% of hypermutator AYAs ($n = 314$, Fig. 4c). Applying the same criterion identified a dominant signature in 93% of OA hypermutators ($n = 5229$). These signatures reflected various molecular defects and mutagen exposures, including DNA mismatch

repair deficiency (MMR, $n = 170$), ultraviolet exposure (UV, $n = 38$), POLE mutation ($n = 19$), APOBEC ($n = 25$), smoking/tobacco ($n = 23$), temozolomide (TMZ) exposure (TMZ, $n = 27$) and BRCA1/2 mutations ($n = 12$).

We found signatures caused by genetic defects were more frequent in AYAs than in OAs (Fig. 4d, Supplementary Data 8), including signatures caused by BRCA1/2 mutations (3.8% vs. 1.8%), MMR deficiency (54.1% vs. 36.9%), and POLE/D mutations (6.1% vs. 2.7%). In contrast, signatures associated with environmental factors were more frequent in OAs, including the APOBEC signature (8.0% vs. 15.5%), smoking/tobacco (7.3% vs. 18.1%) and UV exposure (12.1% vs. 21.0%). The higher proportion of tumors with the TMZ signature (8.6% vs. 4.1%) in AYAs was due to glioma. Temozolomide is a standard chemotherapy agent in the treatment of gliomas, and younger patients typically have better tolerance for TMZ and thus receive higher overall doses.

Some signatures exhibited cancer preferences in AYAs (Fig. 4c). Most of the AYA tumors (84%) carrying the UV signature were melanomas, suggesting excessive sun exposure is a major risk for AYAs. Interestingly, AYA tumors with the smoking/tobacco signature, another mutagen-related signature, were not predominantly found in lung cancer. Instead, we observed the signature in diverse cancer types, suggesting exposure to tobacco may exert widespread mutagenic effects in AYAs. AYA tumors with the POLE signature were mainly colorectal (63%) and endometrial cancers (26%). More than half of the tumors (52%) carrying the APOBEC signature were BRCAs. The TMZ signature was mainly found in gliomas, consistent with the fact that TMZ was part of standard of care for gliomas. Compared to other glioma hypermutators, those carrying the TMZ signature were more likely to be recurrent, likely due to prolonged use of the chemo-agent[37].

The MMR deficiency signature accounted for 54% of all AYA hypermutators and was observed in 21 cancer types. These tumors also exhibited more indels compared to other hypermutators (median indel rate 6.08 vs. 0.85, $p < 2.2e{-}16$, Wilcoxon rank sum test). The most frequent cancer type that carried this signature was colorectal cancer ($n = 89$), followed by glioma ($n = 34$), mature B-cell cancer ($n = 9$), endometrial cancer ($n = 6$), and BRCA ($n = 5$). The finding of the MMR signature in 34 gliomas, mostly glioblastoma ($n = 21$), was surprising because microsatellite instability (MSI) is rare in glioma. However, in a recent study, Touat et al. showed that MSI related short insertion/deletion (INDELs) can be detected by single cell analysis but would be missed by bulk sample analysis[38]. Thus, high depth panel sequencing data can provide sufficient coverage to detect even subclonal INDELs. The indel-to-snv ratios were lower in MMR gliomas than in other MMR tumors ($p = 4.3e{-}6$, Wilcoxon rank sum test), but still higher than those in gliomas exhibiting the TMZ signature ($p = 9.4e{-}3$, Fig. 4e).

Not all hypermutators respond to ICB despite high TMBs. To associate AYA hypermutator signatures with responses to ICB, we mapped the hypermutators to the MSK immunotherapy cohort ($n = 1647$). We identified 15 AYA hypermutators who received PD-L1 or PD1/PD-L1 combinatorial therapy, of which 10 were alive at the time of last follow-up. Seven of the 10 cases where patients were surviving exhibited the MMR signature. In contrast, only one of the five deceased carried the MMR signature, and this patient showed the longest overall survival compared with others. Though we did not have response data and the cohort size was small, these observations suggest that the MMR signature may predict better responses to ICB in AYAs. Thus, given the diversity of cancer lineages where the MMR signature was identified, our data suggest it may be beneficial to expand MSI testing for AYAs in cancer types in which this test is not routinely offered.

## Clinical actionability in AYAs

Finally, we sought to provide a panoramic view of actionable mutations in AYAs. We annotated each mutation with confidence

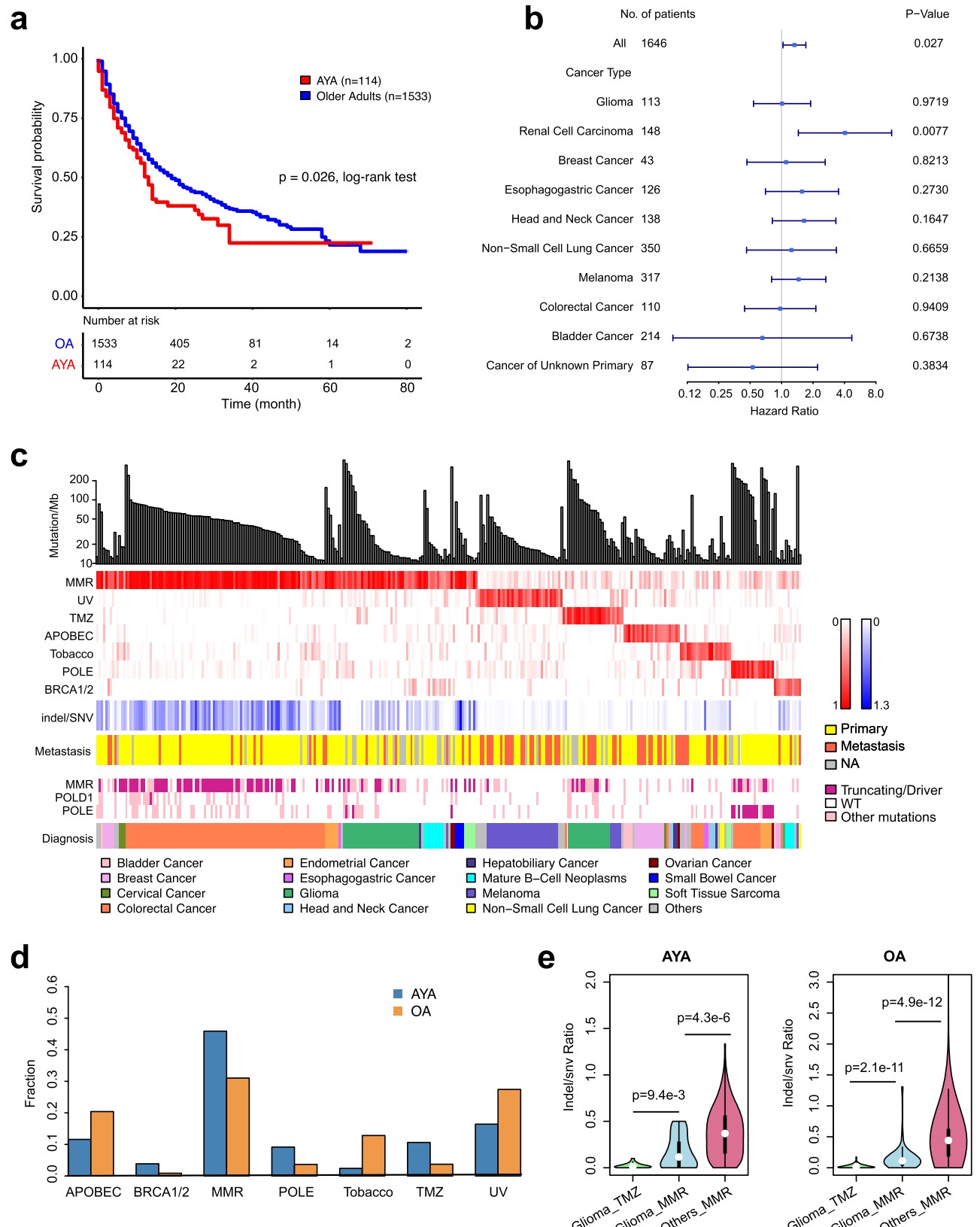

tiers of actionability according to OncoKB[39], a manually curated database for actionable alterations and targeting drugs (Fig. 5). This mapping suggested that 11.8% (1079 of 9140) of tumors in AYAs harbored a level 1 or 2 alteration, 2.3% ($n = 213$) harbored a level 3A alteration, and 23.5% harbored a level 3B alteration ($n = 2147$). Per OncoKB definition, level 1 alterations are those that

can be targeted with an FDA-approved drug; level 2 are standard care biomarkers that can predict response to an FDA-approved drug; level 3A are alterations that are likely targetable by an investigational drug, and level 3B are alterations that are targetable by an approved or investigational drug from a different cancer type.

**Fig. 4 | Hypermutators in AYAs. a** Kaplan–Meier curves showing overall survival of patients who received immune checkpoint inhibitors (AYA, $n = 114$; OA, $n = 1533$; $p = 0.026$, log-rank test). **b** The forest plot shows the Hazard Ratios (HRs) of AYA status on overall survival in ICI cohorts as a whole and in individual cancer types. The error bars represent the 95% confidence interval. $P$ values on the right side of the plot are derived from two-sided log-rank test. **c** Hypermutators in AYAs. From top to bottom are TMB, mutational signatures, indel/snv ratio, metastasis status (orange, metastasis; yellow, primary; gray, unknown), mutation status of DNA mismatch repair genes and *POLE/POLD* genes, and tumor types. For *POLE/POLD* mutations, red indicates driver mutations, pink represent other nonsynonymous

mutations. For MMR genes (*MSH2, MSH6, MSH3, PMS2, MLH1* and *MLH3*), red represents truncating mutations and pink represents all other nonsynonymous mutations. **d** Relative strength of the seven hypermutator-associated mutational signatures in AYAs and OAs. **e** Indel/snv ratios between glioma with the TMZ signature (AYA, $n = 19$; OA, $n = 94$), glioma with the MMR signature (AYA, $n = 34$; OA, $n = 40$) and other cancer types with the MMR signature (AYA, $n = 136$; OA, $n = 1891$). Statistical differences were assessed using two-sided Wilcoxon rank sum test. Center white dot represents the median; the thick black bar in the center represents the upper and lower quartiles; whiskers represent the 1.5 interquartile range. Violin width represents data density. Source data are provided as a Source Data file.

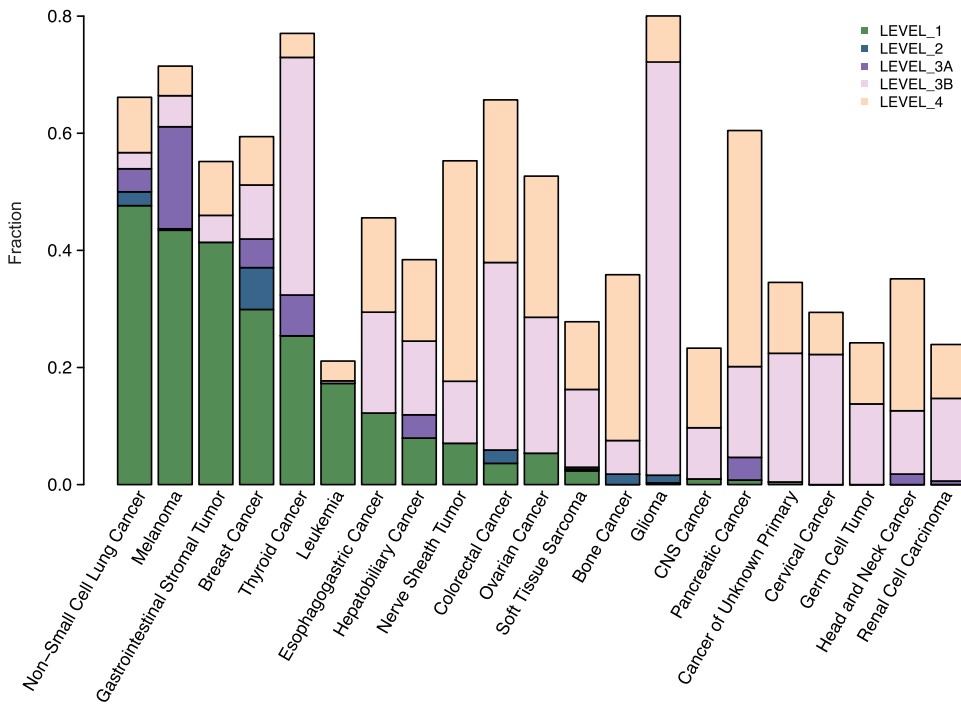

**Fig. 5 | Landscape of clinical actionability in AYAs.** Tumor types are shown in descending order based on the overall frequency of level 1/2 actionable alterations. Actionability was obtained from OncoKB. For each sample, the highest level of actionability of any variant is considered if multiple actional mutations are identified. Only tumor types with more than 80 AYA samples were included in this plot. Source data are provided as a Source Data file.

Specifically, we observed at least one level 1/2 actionable alteration in 50.0% of non-small cell lung cancer, 43.7% of melanoma, 41.4% of gastrointestinal stromal cancer, 37.0% of BRCA, and 25.4% of THCA, among other cancer types that showed a lower proportion of samples with such actionable alterations. The actionable alterations also differed across cancer types. In non-small cell lung cancer, 46.5% of level 1/2 actional alterations were *EGFR* mutations, followed by *ALK* fusions (31.5%) and *ROS1* fusions (8.7%); in melanoma were *BRAF* V600 mutation (98.8%%); in gastrointestinal stromal were *KIT* mutations (88.9%) and *PDGFRA* mutations (11.1%); in BRCA were *ERBB2* amplification (44.4%) and *PIK3CA* mutations (43.7%); in THCA were *RET* mutations or fusions (83.9%) and *NTRK* fusions (16.1%).

Melanoma had the highest proportion of patients with level 3A alterations (17.4%). Glioma had the highest proportions of patients with level 3B alterations (70.6%), most of which (75.0%) were *IDH* R132 mutations that can be targeted with Ivosidenib, a drug approved for treating acute myeloid leukemia and cholangiocarcinoma. In contrast, tumor types such as germ cell tumor, CESC, renal cell carcinoma and head and neck cancer harbored no actionable alterations.

## Discussion

The genomic landscape of AYA cancers is understudied, in part because AYA cases are poorly represented in either adult or pediatric

cancer projects. Meanwhile, cancer incidences are rising among young adults including AYAs. Studies associating age with cancer genomics have been largely focused on comparing early- and late-onset tumors[12–15]. However, these comparisons miss the AYA group, leaving a knowledge gap.

In this work, we systematically investigate the clinical and genomic disparities between AYA and OA cancers. We found substantial differences in clinical presentation between the two age groups, including patient sex, ethnicity, metastatic status, overall survival, and disease histological subtypes. AYA cancers generally show lower mutational load and GI. In accordance, most cancer genes show less mutations and copy number alterations in AYAs. However, we did observe exceptions such as higher GI in AYA BRCAs. Further, analyses show clinical factors including histological subtypes and tumor metastatic status can affect the genomic disparity. We further show that AYA cancers generally harbor more fusions but less TERTp mutations. Analyses of hypermutators suggest AYAs are more susceptible to endogenous mutagenic processes such as DNA mismatch repair deficiency. In comparison, exogenous processes such as UV and tobacco exposure exert greater impact on OAs, suggesting differential etiology for AYA and OA cancers.

We found patterns both consistent and inconsistent with previous reports. For instance, we confirmed the increased mutation rates of

*BRAF* in AYA melanoma reported by Wilmott et al.[40]. Previous studies also suggested more triple-negative subtype in AYA BRCA; however, we did not observe this difference using the TCGA data. Further, earlier studies suggested AYA cancers are more often diagnosed at advanced stages. We only found this pattern in non-small cell lung and pancreatic cancer. Among other six cancer types in which metastatic diseases were differentially diagnosed, AYA patients presented with less metastatic diseases, suggesting improvement in early diagnosis in AYAs.

The high mutation rates of epigenetic regulators such as *H3F3A*, *KDM5A*, and *EP300* suggest a unique etiologic aspect of AYA cancers that remains poorly understood. In addition to mutations in epigenetic regulators, we also observed genomic variants that could contribute to epigenetic changes are more common in AYAs. An example is *SDH* genes mutation. *SDHA* mutation rate was higher in glioma and gastrointestinal stromal tumor (GIST) AYAs. We also identified higher ratio of the pseudohypoxia subtype, which enriched for *SDHA/SDHB* mutations in TCGA paraganglioma and pheochromocytoma AYAs. Previous studies found elevated hypermethylation in SDH-deficient GISTs compared with SDH wildtype GISTs. A similar proportion of changes were found in SDH-deficient paraganglioma and pheochromocytoma, as well as IDH-mutant glioma[41]. Further study shows these DNA methylation changes disrupted the boundary between enhancer and oncogene, strongly upregulated expression of some targetable oncogenes[42]. Those observations and studies suggest targeting consequent aberrant pathways as treatment strategies in epigenetic altered AYA tumors.

In June 2020, The Food and Drug Administration (FDA) approved pembrolizumab, a programmed death 1 (PD-1) blocking antibody, for treating refractory cancers with a TMB >10 mutations per megabase. Therefore, we investigated hypermutators in AYAs. Interestingly, glioma is the second cancer type, following colorectal cancer, with most MMR cases. Although hypermutant adult brain tumors were considered 'immunologically cold' and were unresponsive to ICI[38,43]. Positive cases were reported in glioma with MMR-deficiency[44]. A recent study suggests the PDL1 inhibition is effective in childhood cancers with germline MMR gene mutation[45]. Together, these results indicate that AYA gliomas with MMR-deficiency signature but not TMZ signature could potentially benefit from PD-1 based immunotherapy.

One challenge with AYA and OA genetic comparisons was to control for confounding factors. Such factors can arise from clinical and molecular characteristics that affect tumor mutation profiles. For instance, mutations in metastatic tumors may be affected by treatment and the evolutionary pressure imposed on tumor cells during seeding. Common mutation causing mechanisms, such as ageing, smoking, and UV exposure can also affect gene mutation rate. In our model, we controlled for cancer histology, metastatic status, and patient sex because of the differences exhibited between AYAs and OAs. We also tested clinical and molecular variables including tumor stage, TMB, patient race and ethnicity, and contributing sites. However, more should be tested when such data are made available. Importantly, some variables are relevant to one cancer type but not others. Such variables should be accounted in cancer type specific models.

Despite the advantage to control for confounders by multivariable regression models, univariable models are useful and necessary because determining the most relevant, or an exhaustive list of confounders is difficult, especially in a pan-cancer context. Furthermore, adding a variable to the multivariable model can reduce its statistical power for detecting AYA effect. Univariable models use the full GENIE dataset, thus providing a conservative, reliable method to detect genomic disparities between AYAs and OAs. From a clinical perspective, overrepresentation of a mutation in AYAs can inform basket trials and molecular diagnosis regardless of other clinical and molecular parameters.

An area to further our understanding of the differences between AYAs and OAs is germline mutations, which we did not compare due to data availability. Germline mutations may disproportionately affect younger cancer patients. For instance, approximately half of AYA women with BRCA under the age of 30 harbor a deleterious germline mutation in *BRCA1*, *BRCA2*, or *TP53*[46]. Germline *TP53* mutations cause Li-Fraumeni syndrome (LFS), and *ERBB2* amplification is enriched in LFS-associated BRCA[47–49]. Interestingly, we also observed enriched *ERBB2* amplification in AYAs, although it is unclear if this enrichment is associated with LFS. Nevertheless, the concurrent enrichment of *TP53* mutations and *ERBB2* amplification in AYAs indicate functional cooperation between the oncogenic alterations in driving AYA BRCA.

In summary, we analyzed genomic features of pan-AYA cancers using panel sequencing data. Future efforts to understand the cooperation between germline mutations and somatic mutations in AYA cancer will facilitate a hybrid approach combining genetic counseling and multimodal treatments to reduce the mortality and morbidity caused by cancer in AYAs.

## Methods

### Data sources

Somatic mutations and clinical data were downloaded from the AACR GENIE project via Synapse (release 9). This study was performed in strict accordance with the recommendations of data access guideline of AACR project GENIE datasets. In total, this release included panel sequencing data from 112,935 samples. We used 'AGE_AT_SEQ_REPORT' to determine AYA (15–39 years old) and OA groups (>39 years old), assuming the time of clinical sequencing is close to age at the time of diagnosis because of the fast turnaround for clinical sequencing. We removed samples with unknown ages ('Unknown', $n = 6215$) and pediatric samples ('<18', $n = 3785$), which are mostly gliomas.

We used the MSK-IMPACT cohort to examine differences in overall survival after immune checkpoint inhibitor therapy between AYA and OA groups. Among the 1661 patients who were treated with Immune checkpoint inhibitor (ICI), 114 were AYAs and 1533 were OAs. We used TCGA data to compare molecular subtypes. TCGA molecular subtype data and clinical data were downloaded from PanCanAtlas (https://gdc.cancer.gov/about-data/publications/).

### Clinical and demographic feature comparison

Only cancer types with >100 samples in both AYA and OA groups were retained for clinical and demographic feature comparisons. This filter resulted in 7579 AYA cases and 72,491 OA cases across 19 cancer types. We compared sex, self-reported race and ethnicity (non-Hispanic White, non-Hispanic Black, Hispanic, non-Hispanic Asian), sample type (primary or metastatic) and histologic subtypes using Fisher's exact test, followed by Benjamini–Hochberg method for multiple hypothesis testing.

### TERTp mutation

We extracted 6553 TERTp mutations from the MAF file (data_mutations_extended.txt from GENIE v9.0). Then we limited our analyses to two hotspot promoter mutations at positions 1295250 and 1295228 ($n = 5522$). Those mutations were called from 12 assays. For TERTp mutation comparison, we only used samples from these 12 panels (AYA, $n = 4650$; OA, $n = 49,597$). For each cancer type, we required at least 50 samples in both AYA/OA groups and more than 5 mutant samples in each group. These sample-size prerequisites left us with 15 cancer types to analyze.

### GI score comparison

We calculated GI score using DNA copy-number segmentation data (file 'genie_data_cna_hg19.seg'). Specifically, for each sample, we considered segments to be amplified or deleted if their copy number values were >0.2 in the corresponding direction. GI scores were calculated as the ratio of the total length of these segments over the total length of all segments. In total, we calculated GI score for

53,325 samples and compared AYAs and OAs in 16 cancer types where each group had ≥50 samples.

## Single gene test strategy for mutation rate comparison

To compare somatic mutations between AYAs and OAs that were sequenced with different gene panels, we adopted a 'single gene test' strategy. We note that some panels only cover hotspot regions of certain genes. These would create problems when being analyzed together with other panels that cover the whole exonic regions of the gene. To ensure fair comparison, we removed these "hotspot" panels from our analysis. We further removed all panels from Wake Forest because of their incomplete coverage information in this release. In total, we curated 1029 genes and panels, including 6657 AYA samples and 67,767 OA samples. For a gene-cancer pair to be considered in the comparison, the cancer type must have at least 20 samples in both age groups and more than 5 mutants in either AYAs or OAs. The relatively lenient cohort size requirement was to ensure the inclusion of more genes. Only non-synonymous coding mutations and indels were considered.

Mutation rates were also compared between AYAs and OAs using multivariable logistic regression analysis (AYA status as one variable) adjusting for patient sex, histological subtype and metastatic status. For histological subtype, we only consider major subtypes with more than 15 samples. The rest were grouped as 'others'. In the model all covariates were used as fixed effects and the outcome variable was the mutation status of the examined gene in a cancer type.

## Univariable vs. multivariable models

We used both univariable and multivariable regression models to compare AYAs and OAs in this study. The univariable Fisher's test identified 79 genes, and the multivariable regression model identified 50, 41 overlapped between the two lists. To demonstrate the difference in statistical power of the models, we compared the 38 genes identified by Fisher's test but not the logistic model, and the remaining 41 genes that were identified by both models. The average mutation rate of these 38 genes in AYA cohorts was 9%, compared to 15% for the 41 genes. The average number of mutants of these 38 genes in AYAs was 22, compared to 79 of the other 41 genes. The total number of AYAs used in the analysis was 12,714 for the 38 genes, compared to 20,908 for the other 41 genes. These numbers show that the univariable model is more sensitive when sample size is smaller.

## Controlling for confounders

In our multivariable regression model, we controlled for confounders including cancer histology, metastatic/primary status, and sex. However, other clinical and molecular variables may also confound the analysis. These variables include common clinical features such as cancer stage/grade and patient race and ethnicity, disease specific parameters such as alcohol consumption in liver cancer, and molecular parameters such as tumor mutational burden (TMB). In addition, GENIE data are contributed by different institutions; therefore, sample site can be another confounder. Below we describe how we determined the impact of a confounder. Briefly, for each confounder, we constructed a model with and without it, and then we compared the AYA effect size between the two models. The GENIE provided basic clinical information for tumor samples (age, ethnicity, and diagnosis), but many clinical parameters such as cancer stage/grade, smoking history, alcohol consumption, were not available. We thus could not test them. Some of these can be inferred with mutational signature analysis, but we reasoned that the uncertainties associated with the inferences could negatively affect model credibility.

**Tumor stage.** We used the recently published MSK_MET dataset[50] to test the effect of including tumor stage in multivariable logistic model. MSK_MET consists of 28,789 samples in total, most of which are metastatic samples. Though the dataset has more clinical annotations, tumor stage and grade were still not included. However, based on the clinical information provided and conventional staging criteria, we classified tumor stages as follows:

(1)  tumors without metastasis were classified as 'Stage I' ($n = 2806$)
(2)  tumors with metastasis to regional or distant lymph nodes were classified as 'Stage II/III' ($n = 641$).
(3)  tumors with metastasis to other organs as 'Stage IV' ($n = 24,572$).

Next, we used this dataset to associate tumor stage with AYA genomic disparities. We built two logistic regression models,

1.  mutation status ~ aya status + histological subtypes + metastasis/primary + sex
2.  mutation status ~ aya status + histological subtypes + tumor stage + sex

Note that in the second model where tumor stage was included, the "metastasis/primary" variable was removed because the two were dependent.

With model 1, we detected 10 gene-cancer pairs that were significantly associated with AYA status (FDR < 0.05, Supplementary Data 9). With model 2, we detected 9, all among the 10 detected in model 1 (Supplementary Data 9). The only exception was *CARD11* in BRCA, which was marginally significant in model 2 (FDR = 0.08). These results show high consistency between the two models, likely because the variable metastasis/primary in the original model largely retains the variability of tumor stage.

**Tumor mutational load (TMB).** To explore the impact of TMB on the model, we built a second model including TMB as a variable (mutation~AYA + histology + sex + metastasis + TMB) and compared its output with the original model (mutation ~ AYA + histology + sex + metastasis). For brevity, we call them the TMB model and the original model. We applied the two models to 1025 genes from 16 assays across 81,025 samples. These 16 assays have larger panels thus their TMB estimation is more accurate. With the initial model, we identified 50 significant genes (FDR < 0.05). With the second model, we identified 75 significant genes (FDR < 0.05), 40 of which overlapped with the 50 genes identified in the initial model.

In Supplementary Fig. 2a, we plot the effect size of the AYA variable from the two models for the significant genes. The two sets of effect sizes were highly correlated (rho = 0.96, $p < 2.2e-16$. Gray line is y = x), suggesting the two models give highly similar results. However, most of the significant genes reported solely by the TMB model generally show positive effect size. This observation suggests the TMB model is more sensitive at detecting AYA enriched mutations.

Specifically, 30 of the 35 unique genes identified in the TMB model showed higher mutation rates in AYAs compared with OAs. However, we noticed 23 of the 30 AYA enriched genes were mutated at very low frequencies (on average, 5%; see also Supplementary Data 3), in no more than 5 AYA tumors. Their AYA effect sizes showed much higher standard error compared with other genes (0.7 vs. 0.3), likely due to the small sample size used for the estimation. Given these observations, we were not convinced these 23 genes were robust because mutation status change of a single case can drastically influence the gene's overall effect size. In the original model, we also identified genes ($n = 3$) that were mutated in no more than 5 AYA tumors, but all of them showed lower mutation rates in AYAs than in OAs. The other 7 genes with higher mutation rates from the TMB model appear credible. They included *FANCM*, *FANCD2*, and *BRCA1* in BRCA, *TP53* in Appendiceal Cancer, *BRAF* in melanoma, *GLI3* in non-small cell lung cancer, and *SDHA* in GIST. Several of them (*SDHA*, *FANCM*, *TP53*, *BRAF*) were barely missed by the original model, with

FDR at 0.06, 0.07, 0.07, and 0.051. The identification of *FANCM*, *FANCD2*, and *BRCA1* in BRCA reiterates the role of DNA repair deficiency in AYA BRCA.

Ten genes identified by the initial model were no longer significant in the TMB model. Six of the ten genes showed a lower mutation rate in AYA tumors, all found in melanoma (Supplementary Data 3). This observation makes sense because among all cancer types analyzed, melanoma shows the most pronounced difference in TMB between AYAs and OAs, and thus, accounting for TMB generated a noticeable impact on the results in this cancer type. We note that TMB differences in melanoma between AYAs and OAs are likely mainly driven by UV exposure but less so by age. This explains why accounting for TMB generated less impacts on other cancer types.

Analysis of the other four genes detected by the original model but not the TMB model provides further insight into TMB adjustment in the model. These four genes all showed higher mutation rates in AYAs, including *PTCH1* in embryonal tumor, *KDM5A* in glioma, *MSH2* in colorectal cancer and *WT1* in leukemia. *MSH2* is a DNA repair gene; its mutation causes MSI and high TMB in colorectal cancer. We think exclusion of *MSH2* by the TMB model is unfounded, because *MSH2* mutations are the cause rather than consequence of high TMB. Both *PTCH1* and *WT1* are essential for early development, and both were previously associated with younger patient age[51,52]. These data support the validity of their higher mutation rates in AYAs.

In summary, these results show the value and limitation of including TMB in the modelling. On the one hand, adjusting for TMB can indeed remove effects associated with overall higher TMB in OAs such as in melanoma. On the other hand, it can make the model overly sensitive toward genes with relatively higher mutation rates in AYAs even when their mutation rates are low. It also does not differentiate closely associated gene effects and TMB effects such as *MSH2* and MSI.

**Race and ethnicity.** We also constructed a second model by including race and ethnicity in the model (white_nonHispanic, Black_nonHispanic, Asian_nonHispanic and others, the reference was White_nonHispanic because of the largest sample size). This model identified 44 significant genes, 41 of which overlapped with the 50 genes identified in the initial model. The other three showed borderline significance in the initial model (FDR 0.05–0.08, Supplementary Data 3). Similarly, 9 genes that identified in initial mode but not in the second model also showed borderline significance in the second model (FDR 0.05–0.09).

We compared the effect size of the AYA status variable estimated from the two logistic models (Supplementary Fig. 2b). Again, they were nearly perfectly aligned on the diagonal line (rho = 1, $p < 2.2e-16$), indicating that race and ethnicity has a marginal impact on model performance.

**Site.** Since GENIE project involved multiple cancer center or institutions, to evaluate the effect of contributing sites, we constructed a second model by including sites (the reference sites were chosen based on the one with largest sample size for the testing gene-cancer pair). In the original logistic model, we identified 50 significant genes (FDR < 0.05). With the second model, we identified 58 significant genes, 50 of which were previously identified in the initial model (Supplementary Data 3). The 8 genes identified in the second model showed borderline significance in the initial model (FDR, 0.05–0.09).

We also compared the effect size of the AYA status estimated from the two logistic regression models. The two estimates were nearly perfectly aligned on y = x line (rho = 1, $p < 2.2e-16$; Supplementary Fig. 2c), suggesting site is not a significant confounder.

**Metastasis.** Metastatic tumors accounted for more than 40% of the GENIE cohort. Because metastatic tumors can have distinct mutational profiles from primary tumors and in some cancer types AYAs and OAs showed different proportions of metastatic samples, tumor metastatic status could confound the comparison between AYAs and OAs. One approach to mitigate this confounding effect was to use primary tumors only, but doing so would lose more than 40% of the GENIE sample size. This loss can particularly affect AYAs in the analysis as they only accounted for ~10% of the total cohort size. In our multivariable regression model, we have controlled for tumor metastatic status. To test if our model can properly adjust for metastasis effect, we constructed a model using primary tumors only. We found the AYA effect size was highly similar between the second model and our original model (rho = 0.90, $p < 2.2e-16$; Supplementary Fig. 3), even though the two models were applied to two different datasets (full GENIE vs. primaries only). Two genes showed relatively large variation in AYA effect size, *KEAP1* in 'cancer of unknown primary' and *WT1* in leukemia. For *KEAP1*, the tumors from the two datasets were likely incomparable because of the ambiguous cancer histology. For *WT1*, the sample sizes were three times different (total, 417 vs. 135; AYA, 56 vs. 23). The mutation rate in AYAs was 21% based on the full GENIE dataset but was only 4% based on primary tumors. We therefore concluded that the original model could control for the metastasis effect.

## Gene level CNA comparison
We used file 'data_CNA.txt' from GENIE v9.0. It contains 934 gene level copy number changes from 81,414 samples. Like GISTIC, gene level copy number values were discretized into −2 (deep deletion), −1 (shallow deletion), 0 (neural), 1 (gain), and 2 (amplification). We used −2 for deletions and 2 for amplifications. For each gene, we excluded samples with NA in the file. Similar to mutation analysis, only cancer types that have ≥ 20 samples in either group were considered. Further, only genes with more than 5 deletion/amplification events in either group were included in the comparison. Fisher's exact test and multivariable logistic regression model were used to identify differential CNAs between AYAs and OAs.

## Fusion
Fusions were downloaded from the file 'data_fusion.txt' from GENIE v9.0. We obtained 25,421 unique fusions derived from 21 assays and 17,529 samples. MSK-IMPACT468 panel called the most fusions (n = 9564, from 7388 samples), followed by DFCI-ONCOPANEL-3.1 (n = 4241). To avoid technical batch effects due to design differences, we only used data from the MSK-IMPACT468 panel.

For fusion frequency comparison, the minimum sample size for AYAs or OAs in each cancer type is 20 cases. We focused on recurrent fusions (frequency >1% in a cancer type) and recurrent fusion genes (frequency >2% in a cancer type) since most fusions are only found in single cases.

We used TCGA data to evaluate transcriptional consequences of fusions enriched in AYA samples. We detected five fusions that were enriched in AYAs. Among them, three were detected in TCGA[53], including RET-NCOA4 in THCA, DNAJB1-PRKACA in LIHC, and EML4-ALK in LUAD. Level3 gene level normalized RSEM expression matrix for each cancer type were downloaded from FIREHOSE.

## TMB calculation and comparison
For similar reasons mentioned in mutation analysis, we removed hotspot assays and assays from Wake Forest. We further excluded panels with <0.9 Mb coverage because smaller panels are not as reliable as bigger panels in estimating tumor mutation load (TMB). A total of 16 assays passed these filters. They comprised 6540 AYA samples and 65,431 OA samples. TMB was calculated as the total number of nonsynonymous mutations divided by the length of the

total exonic target region captured by the assay. Information about assay capture regions was obtained from the file genomic_information.txt.

## Mutational signatures

Most panel sequencing samples do not have enough somatic mutations for deconvolution of mutational signature. We instead focused on hypermutators. To identify these samples, we examined the overall TMB distributions in AYAs and OAs to determine a TMB threshold. Using the following formula: median (TMB) + 2*IQR(TMB), where IQR is interquartile range, we identified 340 (cutoff 10.83 non-synonymous mutations/Mb) hypermutators in AYAs and 5632 (cutoff 14.56 non-synonymous mutations/Mb) hypermutators in OAs.

We used deconstructSigs[36] to determine the weights of previously reported hypermutation related mutational signatures in COSMIC (version 2, March 2015), including 13 major signatures: APOBEC (Signatures 2 and 13); Smoking/tobacco chewing (Signature 4 and 29), BRCA1/2 (Signature 3); MMR (Signatures 6, 15, 20, 21 and 26); UV (Signature 7); POLE (Signature 10) and TMZ (Signature 11). A signature was considered dominant in a sample if >40% of observed mutations were attributable to that signature.

For each sample, we also identified somatic mutations in POLE/POLD and mutations in MMR pathway genes and calculated the ratio of indels-to-SNVs. POLE/POLD driver mutations were defined according to a previous study[34] (downloaded from https://www.ncbi.nlm.nih.gov/pmc/articles/PMC5849393/bin/NIHMS947874-supplement-Table_S3.xlsx). MMR pathway genes including MSH2, MSH6, MSH3, PMS2, MLH1 and MLH3.

## Clinical assessment and matching to clinical trials

To assess clinical actionability of mutations, we annotated mutations, copy number alterations, and rearrangements (fusions) using OncoKB (http://oncokb.org). Mutations were classified in a tumor type-specific manner according to the level of evidence that the mutation is a predictive biomarker of drug response. Briefly, mutations were classified as level 1 if they are FDA-approved biomarkers, level 2 if they predict response to standard-of-care therapies, or level 3 if they predict response to investigational agents in clinical trials. Levels 2 and 3 were subdivided according to whether the evidence exists for the pertinent tumor type (2A, 3A) or a different tumor type (2B, 3B). Level 4 mutations were those with compelling biological evidence supporting them as a marker of response to a drug. We did not include 'TMB-High' as a predictive biomarker in this study.

## Statistical analysis

All statistical analyses were carried out using R v4.2.0. We used Fisher's exact test to compare clinical and molecular features between AYAs and OAs. Multiple testing correction was done using the Benjamini–Hochberg method. FDR 0.05 was used to report significant results. Multivariable logistic regression models were constructed in mutation and copy number comparisons. In these models, AYA status was one variable. Other variables included patient sex, histological subtype, and metastatic status (primary or metastatic). Survival analysis was performed using log-rank test and Kaplan–Meier curve was used for visualization. Hazard ratios were determined using Cox proportional hazards model.

## Reporting summary

Further information on research design is available in the Nature Portfolio Reporting Summary linked to this article.

## Data availability

Genomic and clinical data were downloaded from the AACR GENIE project via Synapse (release 9, https://www.synapse.org/#!Synapse:syn7222066/wiki/410924). This study was performed in strict accordance with the recommendations of data access guideline of AACR project GENIE datasets. TCGA genomic data (mc3), molecular subtype data and clinical data were downloaded from PanCanAtlas (https://gdc.cancer.gov/about-data/publications/). MSK-IMPACT immune checkpoint inhibitor therapy cohort was downloaded from https://www.nature.com/articles/s41588-018-0312-8#Sec7[33]. The processed data generated in this study are provided in the Supplementary Information/Source Data file. The remaining data are available within the Article, Supplementary Information or Source Data file. Source data are provided with this paper.

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

## Acknowledgements

This work was supported by GCCRI pilot and partly by NIH/NCI cancer center support grant to Mays Cancer Center at UT Health San Antonio (P30CA054174, X.W.). This work was also supported by CPRIT (RR170055 to S.Z.). The authors would like to acknowledge the American Association for Cancer Research and its financial and material support in the development of the AACR Project GENIE registry, as well as members of the consortium for their commitment to data sharing. Interpretations are the responsibility of study authors. The results shown here are in whole or part based on data generated by the TCGA Research Network.

## Author contributions

X.W. conceived the study, analyzed and interpreted data. S.Z. supervised the project, analyzed and interpreted data. AM.L. and PJ.H. interpreted data. X.W. and S.Z. wrote the paper with input from AM.L. and PJ.H. All authors read and approved the paper.

## Competing interests

The authors declare no competing interests.
