## [Peer Review File · Nature Communications]

Genomic Disparities between Cancers in Adolescent and Young Adults and in Older AdultsReviewers' Comments:

Reviewer #1:

Remarks to the Author:

This study focuses on the characteristics of cancers arising in the adolescent/young adult population. The authors Wang et al. describe clinical and molecular differences between AYA and older adult cancers in data from AACR GENIE (>100,000 samples from >=19 cancer types). While this work does address an unmet need in an understudied population, there are some statistical and modeling considerations that must be addressed as they may drastically change the study's findings.

Major

1. The authors have considered several clinical variables, but some fundamental variables remain inadequately modeled. For instance, tumour stage/grade may be associated with both age and genomic characteristics. There are also some cancer type-specific risk factors tumour types of interest such as alcohol consumption (hepatobiliary), tobacco history (lung, liver, pancreas, among others), parity (ovarian, endometrial), HPV (cervical), etc that are confounding in the relationship between age and cancer. I appreciate that many of these variables are not described in the data, but the authors may be able to use some inferences from mutational signatures. The authors should also expand their discussion on confounding factors (lines 485-491) in their discussion.

The authors should take caution when combining primary and metastatic for analysis. Metastatic tumours can have distinct mutational profiles arising from their evolutionary trajectory and influenced by factors such as prior therapy and site of metastasis. The authors indeed detected the influences of primary/metastasis in their modelling for many of the gene-specific effects. They should consider omitting metastatic samples entirely and focusing on primaries.

The authors should generally keep in mind the effects of prior therapy, as AACR GENIE samples may not be treatment-naive.

2. The authors acknowledge that TMB is closely associated with age, which they observe in their analyses of SNVs and genomic instability. It then seems likely that the observed increases in gene-specific mutation frequency in OA tumours may attributed to this natural accumulation of mutations over time. The authors should explore including TMB/genomic instability in their statistical modeling to control for time-associated mutation accumulation.

3. The study identifies differences between AYA and OA, but some findings appear closely related to or mirrored in similar effects in pediatric cancers. For example, the authors find increased TP53 and ATRX mutations in AYA glioma: these genes are also frequently mutated in childhood gliomas. Is then the increased TP53 and ATRX mutations frequency in AYA an extension of a phenomenon that starts in childhood? The authors filtered out 3,785 pediatric samples from analysis - can they use these samples to place AYA findings in greater context?

4. The single gene test strategy combines univariable and multivariable modeling results in parallel. The authors compare results that are significant in one but not the other. But, how can the authors be sure that an AYA effect that changes in statistical significance is not explained by a loss of statistical power due to the addition of multiple other variables?

The inclusion of univariable analysis in general seems redundant due to the better modeling provided by the multivariable models.

5. The authors should investigate possible transcriptional consequences of these genomic alterations-particularly the gene fusions -if possible. TCGA may have the necessary DNA- and RNA-sequencing data.

Minor

The author's single gene test strategy leads to differing n for different sets of genes. The sample size per test should be noted in the supplementary tables supporting the gene-specific results. Moreover, the authors should consider providing the logistic modeling estimates for all predictors (not just aya_status) in supplementary so readers can explore the full model results.

The claim of increased alternative lengthening of telomeres in AYA glioma and thyroid cancer (lines 237-245) is stronger than the evidence merits. The authors should provide additional support for this claim or soften the language.

Do the authors perform multivariable modelling on TMB and genomic instability?

The authors appear to use 'sex' and 'gender' interchangeably when they are not.

In lines 480-482, the authors state "We found substantial differences in clinical presentation between the two age groups, including patient sex, ethnicity, disease stages, outcome, and disease histological subtypes." As 'disease stages' more often refers to specific clinical definitions, the authors should change the language to focus on pri/met. Also, 'outcome' should be changed to refer to the precise type of ICB response analysed in the study.

I am unclear what the cut-off values in Figure 2A refer to.

More labels should be added to the figures to improve interpretability:

- Fig1A, 3B, 3C, legends in-figure
- Fig2B, gene/cancer type pair labels
- Fig4A, table describing number of pts surviving at each timepoint should be presented under the KM curve

Reviewer #2:

Remarks to the Author:

Wang et al. present here a systematic analysis comparing Cancers in Adolescent and Young Adults vs. Older Adults using data from AACR GENIE. While multiple studies using TCGA & ICGC have been published, AACR GENIE represents a well-powered cohort with many more AYA samples. However, specifically given the site heterogeneity of such a cohort, possible confounders need to be carefully accounted for. Benchmarking results with published studies could also strengthen conclusions.

Major comments:

1. Possible confounding of site and race/ethnicity: the authors correct for gene panel/hotspots sequenced by each institution in AACR GENIE.
 - a. However, given the diversity of patient sampling, data processing, treatment, etc at each site in AACR GENIE, the sites need to be corrected for or stratified to ensure no confounding.
 - b. Also given the authors already reported race/ethnicity differences, the multivariate model needs to correct for that.
 - c. Given thorough correction in the multivariate model, the Fisher's results may be considered

Supplemental or disregarded.

2. Actionability section, the writing needs clarification. For example “non-small cell lung cancer (50.0%, EGFR mutation, ALK fusion, ROS1 fusion)” does that mean 50% of the sample has either one of the three? All of three? Or 50% had druggable events including those three?

3. As the authors have cited, analyses using TCGA/ICGC have been published. Be good to list what findings replicated & what didn't.

Minor comments:

1. Fig 2C the grid dots in the background are not aligned? Are they supposed to mean sth?
2. Survival analyses need to correct for confounders too.

REVIEWER COMMENTS

General responses to reviewer comments

We thank the editor and both reviewers for the constructive comments. The main, common criticism from the reviewers is to control for additional confounders in the modeling (reviewer 1, disease characteristics, TMB; reviewer 2, site, ethnicity). Here, we lay out our rationale for the responses to these comments.

As the reviewers are aware, variable selection is a core issue in multivariable modeling, and there are no standard ways toward it. Typically, variables are included in a model based on biological knowledge and statistical consideration. But a model rarely includes all candidate variables because doing so would increase the risk of collinearity, which can make the model unstable. Furthermore, adjusting for a variable requires a certain number of events for that variable in the dataset; this can be challenging for mutations in AYAs because of the low mutation rates in some cancer types.

In our analysis, we controlled for cancer histology, patient sex, and primary/metastasis status. These variables constituted the backbone of the multivariable logistic model. They were included because we detected differences in these variables in the clinical data analysis (Figure 1).

The reviewers made excellent suggestions to add disease characteristics, TMB, data center (site), and ethnicity to the model. In practice, adding a variable to the model would likely result in changes in the list of significant genes. Making sense of such changes, however, is not trivial and may need detailed manual analysis (see response to reviewer 1 comment #2 for example). To address reviewers' comments under a more consistent framework, we have focused on model effect size rather than FDR. Specifically, for each new variable, we constructed a new model by adding the new variable to the backbone model (mutation~AYA+histology+metastasis+sex). Then we compared the effect size of the *AYA variable* between the new model and the backbone model. This effect size, i.e., the coefficient, is interpreted as changes in odds ratio (in log scale) as tumor AYA status changes (OA is the reference). Positive effect size usually corresponds to mutation enrichment in AYAs. Genes with higher effect size are more significant. If adding a variable has a marginal impact on model performance, the effect sizes between the new and backbone models would be similar.

For the revision, we have tested each of the variables suggested by the reviewers using this approach. We found that except for TMB, adding these variables had little impacts on model performance. In the revised manuscript, we have included these analyses and added more discussions on confounders. We have also compared our results with those from TCGA.

We again thank the reviewers for their excellent suggestions that have significantly strengthened the manuscript.

Reviewer #1, expertise in cancer genomics and health disparities (Remarks to the Author):

This study focuses on the characteristics of cancers arising in the adolescent/young adult population. The authors Wang et al. describe clinical and molecular differences between AYA and older adult cancers in data from AACR GENIE (>100,000 samples from >=19 cancer types). While this work does address an unmet need in an understudied population, there are some statistical and modeling considerations that must be addressed as they may drastically change the study's findings.

Major

1. The authors have considered several clinical variables, but some fundamental variables remain inadequately modeled. For instance, tumour stage/grade may be associated with both age and genomic characteristics. There are also some cancer type-specific risk factors tumour types of interest such as alcohol consumption (hepatobiliary), tobacco history (lung, liver, pancreas, among others), parity (ovarian, endometrial), HPV (cervical), etc that are confounding in the relationship between age and cancer. I appreciate that many of these variables are not described in the data, but the authors may be able to use some inferences from mutational signatures. The authors should also expand their discussion on confounding factors (lines 485-491) in their discussion.

We thank the reviewer for pointing out the data limitation and agree with the reviewer that tumor stage/grade may confound the modeling. We also agree that more discussions on confounding factors are warranted. In the revised manuscript, we have included more discussions in the method and discussion sections. We thank the reviewer for suggesting mutation signature analysis to infer disease-specific risk factors. Because GENIE only has mutations of cancer genes, inference of mutation signatures based on the limited mutations can be less reliable. Furthermore, the quantitative strength of mutation signatures in a tumor from deconvolution is not as objective as clinical records. We respectfully suggest that adding these inferences in the modeling can be counterproductive because of the inference uncertainties.

To address the reviewer's suggestion on including tumor stage/grade in the modeling, we contacted both GENIE and Dr. Michael Berger, who chairs MSK IMPACT data governance committee. GENIE confirmed to us that stage is currently not curated for the main GENIE dataset. Dr. Berger told us that tumor stage and grade information is currently being collected manually and sporadically across different disease teams. He pointed us to the newly released MSK_MET dataset (Nguyen et al. Cell, 2022. PMID: 35120664). MSK_MET consists of 28,789 samples in total, most of which are metastatic samples. Though the dataset has more clinical annotations, tumor stage and grade were still not included. However, based on the clinical information provided and conventional staging criteria, we classified tumor stages as follows:

(1) tumors without metastasis were classified as 'Stage I' (n=2,806)

- (2) tumors with metastasis to regional or distant lymph nodes were classified as ‘Stage II/III’ (n=641).
- (3) tumors with metastasis to other organs as ‘Stage IV’ (n=24,572).

Next, we used this dataset to associate tumor stage with AYA genomic disparities. We built two logistic regression models,

- 1) mutation status ~ aya status + histological subtypes + metastasis/primary + sex
- 2) mutation status ~ aya status + histological subtypes + tumor stage + sex

Note that in the second model where tumor stage was included, the “metastasis/primary” variable was removed because the two were dependent.

With model 1, we detected 10 gene-cancer pairs that were significantly associated with AYA status (FDR<0.05, Supplementary table 4). With model 2, we detected 9, all among the 10 detected in model 1 (Supplementary table 4). The only exception was CARD11 in breast cancer, which was marginally significant in model 2 (FDR = 0.08). These results show high consistency between the two models. We think this is because the variable metastasis/primary in the original model largely retains the variability of tumor stage. We have included these results in the revised manuscript.

The authors should take caution when combining primary and metastatic for analysis. Metastatic tumours can have distinct mutational profiles arising from their evolutionary trajectory and influenced by factors such as prior therapy and site of metastasis. The authors indeed detected the influences of primary/metastasis in their modelling for many of the gene-specific effects. They should consider omitting metastatic samples entirely and focusing on primaries.

We agree with the reviewer about the impact of metastasis on mutational profiles. We used the full dataset instead of only primary tumors in our original analysis for three main reasons:

- 1) Unlike TCGA that limited sample procurement to treatment naïve primary tumors (except melanoma), GENIE is a more clinically oriented dataset. We thus think using it can attain more clinically relevant insights.
- 2) Using only primary tumors (n=49,960) would lose more than 40% of the GENIE sample size (n=83,482) in single gene tests. This would significantly reduce the power of AYA analysis because AYA cases account for only ~10% of the total cohort.
- 3) We have included *metastasis* as a variable in the logistic model. This adjusts for its impact without significantly reducing the model’s statistical power. As the reviewer pointed out, the multivariable model indeed detected mutation differences between AYAs and OAs while picking up metastasis related effects.

To better demonstrate the last two points, we conducted the same tests (both Fisher’s exact test and logistic modelling) using only primary tumors as the reviewer suggested. Figure R1 shows how the results compared with our original results: while using primaries identified 5 new hits, using all the samples identified 53 more hits that would otherwise

be missed by using primaries only. This is not surprising, given the much larger sample size afforded by the full GENIE dataset.

Figure R1. Significant hits using the whole GENIE cohort (blue) or using primary tumors only (yellow).

Figure R2. Correlation of AYA effect size in models with and without metastatic samples. The x axis represents effect size from the original model, and the y axis represents the effect size from a model based on only primary tumors. Each dot represents one gene-cancer pair identified from all samples or using primary tumors only.

One concern with the metastatic samples is if including them in the modelling would change the effect size of the AYA variable. To test this, we constructed a new model using only primary tumors. We then compared the effect size of the AYA variable for the significant gene-cancer pairs from this new model with the ones from the original model that was based on the full GENIE dataset. Here the effect size indicates the changes in odds ratio when AYA status changes (please see our general response at the top of the rebuttal letter). As shown in Figure R2, the two sets of effect sizes were highly consistent around the $y=x$ line (the grey line; $\rho=0.90$, $p<2.2e-16$). This is remarkable given that

they were derived from two different datasets (full GENIE v.s. primaries only). The high consistency suggests the original model controls the *metastasis* variable well. We did observe two outliers deviating from the $y=x$ line, KEAP1 in cancer of unknown primary and WT1 in leukemia. Further analysis revealed significant differences in sample size for these two cases. For KEAP1, we had 2364 samples (137 AYAs and 2227 OAs) in the original model but only 275 in primary-only model (20 AYAs and 255 OAs). Similarly for WT1, we had 417 samples in the original model ($n=56$ for AYA, 361 for OA) but only 135 in the new model ($n=23$ for AYA, 112 for OA). More importantly, the designation of the disease entity “cancer of unknown primary” strongly suggests that cancer types in the full GENIE dataset and the primary-only dataset for KEAP1 may not be comparable. For WT1, the concept of “metastasis” does not really apply to leukemia. In OAs, WT1 mutation rates were 6.6% and 6.3% in the two datasets; in AYAs, however, the rates were 21% and 4%. The rate disparity in AYAs suggests that fewer AYAs makes estimates of gene mutation rates less robust in a heterogeneous disease cohort particularly for genes with low mutation rates, causing differences in model results. Taken together, these data show that after adjusting for the *metastasis* variable, the original multivariable model based on the full GENIE dataset yields consistent results with models using primaries only.

In the revised manuscript, we have added these new data (Supplementary Table 3 and 4, new Supplementary Figure 3).

The authors should generally keep in mind the effects of prior therapy, as AACR GENIE samples may not be treatment-naive.

We appreciate the reviewer pointing out this very important characteristic of the dataset. Indeed, we found treatment related mutational signatures. For instance, in glioma hypermutators, we observed the mutational signature of temozolomide, a chemo-agent that is part of the standard of care for treating the disease.

2. The authors acknowledge that TMB is closely associated with age, which they observe in their analyses of SNVs and genomic instability. It then seems likely that the observed increases in gene-specific mutation frequency in OA tumours may be attributed to this natural accumulation of mutations over time. The authors should explore including TMB/genomic instability in their statistical modeling to control for time-associated mutation accumulation.

The reviewer raised an excellent point. To explore the impact of TMB on the model, we have built a new model including TMB as a variable (mutation~AYA+histology+sex+metastasis+TMB) and compared its output with the original model (mutation~AYA+histology+sex+metastasis). For brevity, we call them the TMB model and the original model. We applied the two models to 1,025 genes from 16 assays across 81,025 samples. These 16 assays have larger panels thus their TMB estimation is more accurate. With the initial model, we identified 50 significant genes ($FDR<0.05$). With the new model, we identified 75 significant genes ($FDR<0.05$), 40 of which overlapped with the 50 genes identified in the initial model.

Figure R3. Correlation of AYA effect size between the original model (x axis) and the TMB model (y axis). Each dot represents one gene-cancer pair identified from either model.

In Figure R3, we plot the effect size of the AYA variable from the two models for the significant genes. The two sets of effect sizes were highly correlated ($\rho=0.96$, $p<2.2e-16$). Grey line is $y=x$), suggesting the two models give highly similar results. However, we note that this correlation pattern is different from Figure R2, R5, and R6, because most of the significant genes reported solely by the TMB model generally show positive effect size (yellow dots in Figure R3). This observation suggests the TMB model is more sensitive at detecting AYA enriched mutations. This is consistent with the reviewer's comments that increases in mutation rates in OAs can be partially explained by their higher TMB instead of their age stratum.

Specifically, 30 of the 35 unique genes identified in the TMB model showed higher mutation rates in AYAs compared with OAs. However, we noticed 23 of the 30 AYA enriched genes were mutated at very low frequencies (on average, 5%; see also new Supplementary Table 3), in no more than 5 AYA tumors. Their AYA effect sizes showed much higher standard error compared with other genes (0.7 vs 0.3), likely due to the small sample size used for the estimation. Given these observations, we are not convinced the identification of these 23 genes is robust because mutation status change of a single case can drastically influence the gene's overall effect size. In the original model, we also identified genes ($n=3$) that were mutated in no more than 5 AYA tumors, but all of them showed lower mutation rates in AYAs than in OAs. The other 7 new genes with higher mutation rates from the TMB model appear credible. They included FANCM, FANCD2, and BRCA1 in breast cancer, TP53 in Appendiceal Cancer, BRAF in melanoma, GLI3 in non-small cell lung cancer, and SDHA in gastrointestinal stromal tumor. Several of them (SDHA, FANCM, TP53, BRAF) were barely missed by the original model, with FDR at 0.06, 0.07, 0.07, and 0.051. The identification of FANCM, FANCD2, and BRCA1 in breast cancer reiterates the role of DNA repair deficiency in AYA breast cancer.

Ten genes identified by the initial model were no longer significant in the TMB model. Six of the ten genes showed a lower mutation rate in AYA tumors, all found in melanoma (Supplementary Table 3). In Figure R3, these genes (green dots with negative effect sizes) showed decreased AYA effect size by an average of 42% in the TMB model. This observation makes sense because among all cancer types analyzed, melanoma shows the most pronounced difference in TMB between AYAs and OAs (see Figure 2), and thus, accounting for TMB generated a noticeable impact on the results in this cancer type. We note that TMB differences in melanoma between AYAs and OAs are likely mainly driven by UV exposure but less so by age. This explains why accounting for TMB generated less impacts on other cancer types.

Analysis of the other four genes detected by the original model but not the TMB model provides further insight into TMB adjustment. These four genes all showed higher mutation rates in AYAs, including PTCH1 in embryonal tumor, KDM5A in glioma, MSH2 in colorectal cancer and WT1 in leukemia. MSH2 is a DNA repair gene; its mutation causes microsatellite instability and high TMB in colorectal cancer. We think exclusion of MSH2 by the TMB model is unfounded, because MSH2 mutations are the cause rather than consequence of high TMB. Both PTCH1 and WT1 are essential for early development, and both were previously associated with younger patient age (Guo et al. PMID 24204797; Gaidzik et al. PMID 19221039). Though we do not know why they were not detected by the TMB model, their functions clearly support the validity of their higher mutation rates in AYAs.

In summary, these new results show the value and limitation of including TMB in the modelling. On the one hand, adjusting for TMB can indeed remove effects associated with overall higher TMB in OAs such as in melanoma. On the other hand, it can make the model overly sensitive toward genes with relatively higher mutation rates in AYAs even when their mutation rates are low. It also does not differentiate closely associated gene effects and TMB effects such as MSH2 and microsatellite instability.

In the revised manuscript, we have included outputs from both models in Supplementary Table 3&4). We have also updated our method regarding the above discussions. We thank the reviewer for raising this important question.

3. The study identifies differences between AYA and OA, but some findings appear closely related to or mirrored in similar effects in pediatric cancers. For example, the authors find increased TP53 and ATRX mutations in AYA glioma: these genes are also frequently mutated in childhood gliomas. Is then the increased TP53 and ATRX mutations frequency in AYA an extension of a phenomenon that starts in childhood? The authors filtered out 3,785 pediatric samples from analysis - can they use these samples to place AYA findings in greater context?

We excluded the 3785 samples because GENIE does not provide precise age information for these cases. Instead, their ages are described as "<18" or "unknown." Our concern was that without precise age, we could not possibly determine which cases were AYA and which were pediatric.

Nevertheless, we calculated the mutation rates of highly mutated glioma genes in pediatric (<18 yr), AYA (15-39 yr), and OA (>39 yr) gliomas, acknowledging that there may be some overlaps between pediatric and AYA samples. The results are shown below (Table R1). While BRAF and H3F3A show decreased mutation rates as patient age increases, TP53, ATRX, and IDH1 defy this pattern, with all three showing the highest mutation rate in AYAs. GENIE has few pediatric samples for other cancer types. We have included a brief discussion on these data in the revised manuscript.

	pediatric	AYA	OA
BRAF	0.175	0.074	0.033
TP53	0.175	0.540	0.321
ARTX	0.069	0.413	0.125
IDH1	0.037	0.556	0.188
H3F3A	0.106	0.067	0.008

Table R1. Mutation rates of several glioma related genes in pediatric, AYA and OA cohorts.

4. The single gene test strategy combines univariable and multivariable modeling results in parallel. The authors compare results that are significant in one but not the other. But, how can the authors be sure that an AYA effect that changes in statistical significance is not explained by a loss of statistical power due to the addition of multiple other variables?

The inclusion of univariable analysis in general seems redundant due to the better modeling provided by the multivariable models.

We thank the reviewer's question, which prompted us to examine the univariable and multivariable models more closely. We agree that adding other variables may reduce the multivariable model's statistical power in detecting AYA effects; that is why the univariable model is necessary, because it can use the full sample size to test AYA effects. We also agree with the reviewer about the advantages of multivariable models for their ability to control for confounders. However, we respectfully point out the limitation of the multivariable models: precisely as the reviewer commented, these models lose statistical power when more variables are added. Acknowledging this limitation in our analysis is very important because the AYA cohort size is usually much smaller than that of older adults. Thus, for low mutation rate genes, the limited number of mutation events does not provide the sample size for proper multivariable modeling.

We described in the manuscript that Fisher's test identified 79 significant hits, 38 were not significant in the logistic model. The average mutation rate of these 38 genes in AYA cohorts is 9%, compared to 15% for the 41 genes identified by both Fisher's test and the logistic model. The average number of mutants of these 38 genes in AYAs is 22, compared to 79 of the other 41 genes. The total number of AYAs used in the analysis is 12,714 for the 38 genes, compared to 20,908 for the other 41 genes. These numbers clearly show that when sample size drops, so is the statistical power of the multivariable logistic model.

Another consideration for keeping the univariable analysis is our incomplete understanding of confounding factors. Despite our best effort, there may be other factors that can influence logistic model results, especially in disease specific contexts, as the reviewer insightfully pointed out in comment #1. When a relevant confounder is added or removed from the logistic model, gene list changes. A good example is the identification of BRAF in AYA melanoma when TMB is added to the model (see our response to reviewer comment #2). Fisher's test consistently identified it (FDR=0.012). This consistency is important, because from a clinical perspective, overrepresentation of a gene mutation in AYA patients, regardless of other clinical and molecular parameters, can guide basket trials or inform molecular diagnosis. Therefore, we lean toward keeping the data.

In the revision, we have expanded the reasoning for using multivariable logistical model and univariable model in the analysis. We thank the reviewer again for raising this question that has led to these discussions.

5. The authors should investigate possible transcriptional consequences of these genomic alterations- particularly the gene fusions -if possible. TCGA may have the necessary DNA- and RNA-sequencing data.

We detected 5 fusions that were enriched in AYAs. Among them, three were detected in TCGA (Gao et al. PMID 29617662), including RET-NCOA4 in THCA, DNAJB1-PRKACA in LIHC, and EML4-ALK in LUAD. Figure R4 shows the expression of the oncogenes involved in the fusion between fusion-positive and fusion-negative tumors. As the reviewer suspected, fusions are associated with increased expression of these genes. We have included Figure R4 in the revision (new Supplementary Figure 7).

Figure R4. Functional consequence of fusion transcripts that are enriched in AYAs. The expression data were downloaded from TCGA Pan-Cancer Atlas. Fusion calls were downloaded from Gao et al. *Cell Rep*, 2018. The y axis represents expression level of the oncogene that is involved in the fusion. We also include tumors with other alterations of the oncogene for comparison.

Minor

The author's single gene test strategy leads to differing n for different sets of genes. The

sample size per test should be noted in the supplementary tables supporting the gene-specific results. Moreover, the authors should consider providing the logistic modeling estimates for all predictors (not just aya_status) in supplementary so readers can explore the full model results.

Done. They have been included in the new Supplementary Table 4.

The claim of increased alternative lengthening of telomeres in AYA glioma and thyroid cancer (lines 237-245) is stronger than the evidence merits. The authors should provide additional support for this claim or soften the language.

We have reworded the text to soften the claim.

Do the authors perform multivariable modelling on TMB and genomic instability?

Please see our response to comment #2 and our general response at the top of this rebuttal letter.

The authors appear to use 'sex' and 'gender' interchangeably when they are not.

We have replaced all 'gender' with 'sex'.

In lines 480-482, the authors state "We found substantial differences in clinical presentation between the two age groups, including patient sex, ethnicity, disease stages, outcome, and disease histological subtypes." As 'disease stages' more often refers to specific clinical definitions, the authors should change the language to focus on pri/met. Also, 'outcome' should be changed to refer to the precise type of ICB response analysed in the study.

We have modified the text accordingly.

I am unclear what the cut-off values in Figure 2A refer to.

We have added a sentence in the figure legend.

More labels should be added to the figures to improve interpretability:

- Fig1A, 3B, 3C, legends in-figure
We have add legends in those figures.
- Fig2B, gene/cancer type pair labels

The labels would not be legible if we include them for all the pairs. We have provided the source data for generating this figure in Supplementary Table 3.

- Fig4A, table describing number of pts surviving at each timepoint should be presented under the KM curve

We have updated figures accordingly.

Reviewer #2, expertise in age-related differences in cancer, bioinformatics and genomics (Remarks to the Author):

Wang et al. present here a systematic analysis comparing Cancers in Adolescent and Young Adults vs. Older Adults using data from AACR GENIE. While multiple studies using TCGA & ICGC have been published, AACR GENIE represents a well-powered cohort with many more AYA samples. However, specifically given the site heterogeneity of such a cohort, possible confounders need to be carefully accounted for. Benchmarking results with published studies could also strengthen conclusions.

Major comments:

1. Possible confounding of site and race/ethnicity: the authors correct for gene panel/hotspots sequenced by each institution in AACR GENIE.
a. However, given the diversity of patient sampling, data processing, treatment, etc at each site in AACR GENIE, the sites need to be corrected for or stratified to ensure no confounding.

In the original logistic model, we identified 50 significant genes (FDR<0.05). To evaluate the effect of contributing sites, we constructed a new model by including sites (the reference sites were chosen based on the one with largest sample size for the testing gene-cancer pair). With the new model, we identified 58 significant genes, 50 of which were previously identified in the initial model (Supplementary Table 3). The 8 new genes showed borderline significance in the initial model (FDR, 0.05~0.09).

Figure R5. Correlation of AYA effect size between the original model (x axis) and the model with added "site" variable (y axis). Each dot represents a gene-cancer pair identified in either model.

We also compared the effect size of the *AYA* status estimated from the two logistic regression models (please see our general response at the top of the rebuttal). The two estimates were nearly perfectly aligned on $y=x$ line ($\rho=1$, $p < 2.2e-16$; Figure R5), suggesting site is not a significant confounder. In the revision, we have included Figure R5 (new Supplementary Figure 2)

b. Also given the authors already reported race/ethnicity differences, the multivariate model needs to correct for that.

Like 'site', we constructed a new model by including race/ethnicity in the model (white_nonHispanic, Black_nonHispanic, Asian_nonHispanic and others, the reference race was White_nonH because of the largest sample size). This model identified 44 significant genes, 41 of which overlapped with the 50 genes identified in the initial model. The other three showed borderline significance in the initial model (FDR 0.05 ~ 0.08, new Supplementary Table 3). Similarly, 9 genes that identified in initial mode but not in the new model also showed borderline significance in the new model (FDR 0.05 ~ 0.09).

We also compared the effect size of the *AYA status* variable estimated from the two logistic models. Again, they were nearly perfectly aligned on the diagonal line ($\rho=1$, $p < 2.2e-16$), indicating that race/ethnicity has a marginal impact on model performance. We have included the data and additional discussions in the revision (new Supplementary Table 3).

Figure R6. Correlation of AYA effect size between the original model (x axis) and the model with race/ethnicity variable (y axis). Each dot represents a gene-cancer pair identified in either model.

c. Given thorough correction in the multivariate model, the Fisher's results may be considered Supplemental or disregarded.

We thank the reviewer for the suggestion, which was also raised by Reviewer 1. Please see our response to Reviewer 1 comment #4.

2. Actionability section, the writing needs clarification. For example “non-small cell lung cancer (50.0%, EGFR mutation, ALK fusion, ROS1 fusion)” does that mean 50% of the sample has either one of the three? All of three? Or 50% had druggable events including those three?

We have edited the text to make it clearer.

3. As the authors have cited, analyses using TCGA/ICGC have been published. Be good to list what findings replicated & what didn't.

We thank the reviewer for the suggestion. The four studies we cited in the manuscript identified genomic correlates with age, but in different contexts. Lee et al. (PMID 34788626) compared early-onset (≤ 50 yr) and late on-set tumors (> 50 yr). Shah et al. (PMID 34879281) divided patients based on age distribution and compared patients from the 1st and 4th age quarters in each cancer type. They also divided normal GTEx samples into young (< 50 yr) and old (> 59 yr) in their study. Li et al. (PMID 35017538) and Chatsirisupachai et al. (PMID 33879792) used linear models to find genomic correlates with age. Because TCGA and ICGC samples are heavily biased towards older patients, the linear models had little input from AYA samples. These key differences highlight the novelty of our work, but meanwhile preclude their direct comparison with our data.

In addition to the technical differences, GENIE and TCGA/ICGC cohorts also show clinical differences. Take TCGA for example; TCGA cancer cohorts are often focused on one histology (except sarcomas). For instance, TCGA melanoma primarily consists of cutaneous melanomas, and TCGA thyroid cancer consists of papillary thyroid carcinoma. GENIE is much more diverse. GENIE melanoma consists of cutaneous melanoma, acral melanoma and uveal melanoma. GENIE thyroid cancer consists of papillary thyroid cancer, anaplastic thyroid cancer, medullary thyroid cancer, etc.

Nonetheless, we have tried to validate our results using TCGA data. For TCGA cancer types, we require a minimum of 50 AYA and OA tumors. This led to 5 common cancer types between TCGA and GENIE, including thyroid cancer (THCA), cervical cancer (CESC), breast cancer (BRCA), glioma (GBM/LGG), and melanoma (SKCM). For the genes we identified in the 5 cancer types, we calculated their mutation rate in AYAs and OAs in TCGA, and the differences between the two rates. The difference is one way to measure the effect size for Fisher's test, and it is better than the odds ratio because the latter is sensitive to low mutation rates.

Figure R7 a-b shows the mutation rates of the genes in AYAs and OAs, respectively. The x axis represents the results from GENIE; the y axis represents the results from TCGA; the grey line is $y=x$. We can clearly see a systematical difference between TCGA and GENIE; gene mutation rates are higher in TCGA AYAs than in GENIE, particularly for melanoma. But this difference is not observed in OAs.

Figure R7c shows the OA-AYA mutation rate difference from GENIE (x-axis) and TCGA (y axis). The difference mitigates the systematic difference and shows a consistent pattern between the two datasets. We have included this figure in the revised manuscript (new Figure 2E and new supplementary Figure 4).

Figure R7. Gene mutation rate in TCGA and GENIE. (a) Gene mutation rate in AYAs in GENIE (x axis) and TCGA (y axis). (b) Gene mutation rate in OAs in GENIE (x axis) and TCGA (y axis). (c) Differences in mutation rates between AYAs and OAs based on GENIE (x axis) and TCGA (y axis). Each dot represents one gene-cancer pair that was identified in the logistic model.

Minor comments:

1. Fig 2C the grid dots in the background are not aligned? Are they supposed to mean sth?

We apologize for the confusion. These were due to scaling in Illustrator during manuscript preparation. We have fixed the issue.

2. Survival analyses need to correct for confounders too.

In the revision, we repeated the Cox model analysis by including sex, histology, and metastasis status in the model. We obtained a p value of 0.09 (HR= 1.251, 95% CI: 0.965-1.622). We think this result is consistent with the original results, especially given the sample size of AYA tumors (n=114 across cancer types with outcome data). We have included this data in the revised manuscript.

Reviewers' Comments:

Reviewer #1:

Remarks to the Author:

The authors have addressed my concerns and made necessary revisions to their manuscript. Their methodology is sound and sufficiently transparent so that external parties can easily examine and build on it. This study focuses on a growing population of young adult cancer patients whose diseases are not well understood. The results of this study are of great interest and are a step towards elucidating underlying mechanisms of young adult cancers.

Reviewer #2:

Remarks to the Author:

The authors have satisfactorily addressed my major concerns.

None of the reviewers raised any concern. Below are the reviewers' comments:

Reviewer #1 (Remarks to the Author):

The authors have addressed my concerns and made necessary revisions to their manuscript. Their methodology is sound and sufficiently transparent so that external parties can easily examine and build on it. This study focuses on a growing population of young adult cancer patients whose diseases are not well understood. The results of this study are of great interest and are a step towards elucidating underlying mechanisms of young adult cancers.

Reviewer #2 (Remarks to the Author):

The authors have satisfactorily addressed my major concerns.